# First-Explore, then Exploit: Meta-Learning to Solve Hard Exploration-Exploitation Trade-Offs

**Ben Norman**[1,2]
btnorman@cs.ubc.ca

**Jeff Clune**[1,2,3]
jclune@gmail.com

[1]Department of Computer Science, University of British Columbia
[2]Vector Institute
[3]Canada CIFAR AI Chair

## Abstract

Standard reinforcement learning (RL) agents never intelligently explore like a human (i.e. taking into account complex domain priors and adapting quickly based on previous exploration). Across episodes, RL agents struggle to perform even simple exploration strategies, for example systematic search that avoids exploring the same location multiple times. This poor exploration limits performance on challenging domains. Meta-RL is a potential solution, as unlike standard RL, meta-RL can *learn* to explore, and potentially learn highly complex strategies far beyond those of standard RL, strategies such as experimenting in early episodes to learn new skills, or conducting experiments to learn about the current environment. Traditional meta-RL focuses on the problem of learning to optimally balance exploration and exploitation to maximize the *cumulative reward* of the episode sequence (e.g., aiming to maximize the total wins in a tournament – while also improving as a player). We identify a new challenge with state-of-the-art cumulative-reward meta-RL methods. When optimal behavior requires exploration that sacrifices immediate reward to enable higher subsequent reward, existing state-of-the-art cumulative-reward meta-RL methods become stuck on the local optimum of failing to explore. Our method, First-Explore, overcomes this limitation by learning two policies: one to solely explore, and one to solely exploit. When exploring requires forgoing early-episode reward, First-Explore significantly outperforms existing cumulative meta-RL methods. By identifying and solving the previously unrecognized problem of forgoing reward in early episodes, First-Explore represents a significant step towards developing meta-RL algorithms capable of human-like exploration on a broader range of domains.

## 1 Introduction

Reinforcement learning (RL) [1] can perform challenging tasks, such as plasma control [2], molecule design [3], game playing [4], and robotic control [5]. However, RL is sample inefficient (taking thousands of episodes to learn tasks humans master in a few tries) [6], limiting its application. Meta-RL [7–12] circumvents this issue by enabling an agent to adapt to new environments based solely on prior experience (i.e., remembering what occurred in previous episodes and using that to inform subsequent behavior). By replacing slow weight-based RL updates with memory-based meta-RL adaption, human-like sample efficiency [12] can be achieved.

This paper focuses on *cumulative-reward* meta-RL, which aims to optimize performance across a sequence of episodes, $\tau_1, \ldots, \tau_n$ (e.g., games in a tournament). The objective is to maximize the total reward accumulated over all episodes (e.g., the number of games won), expressed as

38th Conference on Neural Information Processing Systems (NeurIPS 2024).

$\sum_{i=1}^{n} G(\tau_i)$, where the episode return $G(\tau_i)$ is the total reward of episode $\tau_i$. To maximize this sum, the agent should optimally balance exploration and exploitation across the sequence, e.g., prioritizing exploration in early episodes so as to better exploit in later ones.

Cumulative reward meta-RL has an unrecognized failure mode, where state-of-the-art (SOTA) methods achieve low cumulative-reward regardless of how long they are trained. This dynamic occurs in domains with the following properties: **A**. Maximizing the expected total reward requires exploratory actions that forgo immediate reward, and **B**. The benefit of these exploratory actions *only occurs* when they are *reliably* followed by good exploitation (i.e., if exploitation is too inconsistent, then exploration results in *lower* total reward).

An example is a bandit domain where the first arm always provides a reward that is better than the average arm, but not the highest possible. To maximize cumulative reward over many pulls, the agent must explore the other arms, and then repeatedly exploit the best one. Property **A** holds, as this optimal policy forgoes immediate reward by not sampling the first arm and its above-average reward. Property **B** holds, as exploration (sampling arms other than the first) is *only valuable* when it is followed by sufficiently consistent exploitation (reliably re-sampling the best arm).

The failure occurs as follows: 1. At the start of training, the agent, being randomly initialized, lacks the ability to *reliably exploit* learned information. 2. As a result, the domain properties **A** and **B** cause exploratory actions to lead to lower total reward than the other actions do. 3. This lower reward trains the agent to *actively avoid exploration*. 4. This avoidance then locks the agent into poor performance, as it cannot learn effective exploitation without exploration. This process occurs in the bandit example. Initially, the agent cannot exploit (e.g., when it finds the best arm it does not reliably resample it). The associated negative expectation of exploration then trains the agent to *avoid exploration* by only sampling the first arm, with its above-average, but sub-optimal, arm reward.

Current SOTA meta-RL algorithms such as RL$^2$ [8, 9], VariBAD [7], and HyperX [10] attempt to train a single policy to maximize the cumulative reward of the whole episode sequence. This optimization causes step 3 of the above-described failure process, and thus these methods suffer from the issue of failing to properly learn (e.g., converging rapidly to a policy of not exploring), which we demonstrate on multiple domains (Section 6). The issue is especially insidious because distributions of simple environments (each trivially solved by standard-RL) can stymie these methods. Surprisingly, domains such as bandits can be too hard for SOTA meta-RL.

We introduce a new approach, First-Explore (visualized in Figure 1), which overcomes this problem associated with directly optimizing for cumulative reward. Rather than training a single policy to maximize cumulative reward, First-Explore learns two policies: an exploit policy, and an explore policy. The exploit policy maximizes episode return, without attempting to explore. In contrast, the explore policy explores to best inform the exploit policy, without attempting to maximize its own episode return. Only *after training* are the two policies combined to achieve high cumulative reward.

Because the explore policy is trained solely to inform the exploit policy, poor current exploitation no longer causes immediate rewards (property **A**) to *actively discourage* exploration. This change eliminates step 3 of the failure process, and enables First-Explore to perform well in domains where SOTA meta-RL methods fail. By identifying and solving this previously unrecognized issue, First-Explore represents a substantial contribution to meta-RL, paving the way for human-like exploration on a broader range of domains.

## 2  Background

**RL Terminology:** environments are formally defined as partially observable Markov decision processes (POMDPs, [1]). Each POMDP $E$ is specified by a tuple $E = (S, A, p, p_0, R, \Omega, O, \gamma)$, where $S$ is the state space, $A$ the action space, $p : S \times A \rightarrow S$ a probabilistic transition function mapping from the current state and action to the next state, $p_0$ a distribution over starting states, $R : S \times A \rightarrow \mathbb{R}$ a stochastic reward function, $\Omega$ the space of environment observations, $O : S \rightarrow \Omega$ a stochastic function mapping from states to observations, and $\gamma$ the discount factor. The environment starts (at $t = 0$) in a start state $s_0$ according to $p_0$, $s_0 \sim p_0$. Each subsequent time-step, the agent receives the current state's observations $o_i = O(s_i)$, takes an action $a_i$, and the transition function $p$ updates the environment state $s_{i+1} = p(s_i, a_i)$. An episode $\tau$ of length $h$ is then a sequence of time-steps starting from $t = 0$ to $t = h$. The sum of an episode's $\gamma$-discounted rewards is called its

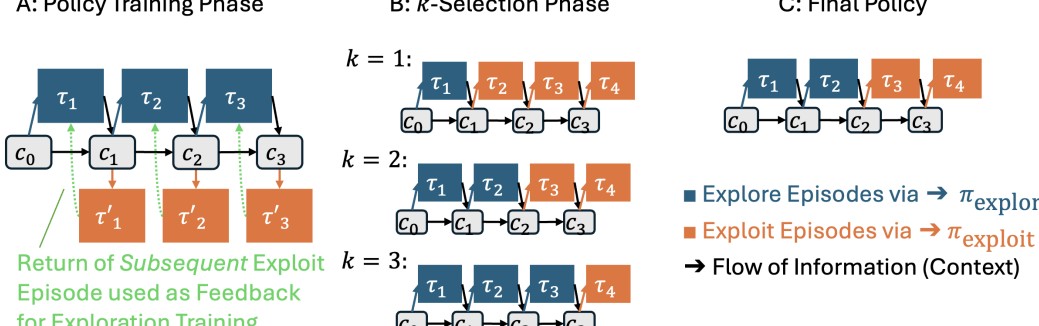

Figure 1: **First-Explore** aims to maximize the cumulative reward of a sequence of $n$ episodes on a target environment distribution. This optimization is achieved by first training two separate policies, and then combining them *after* training to maximize the total reward obtained. **A.** First, two separate policies are trained on the distribution of environments: one to **explore** (produce informative episodes), and one to **exploit** (maximize current episode return). During training, the **explore policy** $\pi_{\text{explore}}$ provides all the context $c_i = \tau_1, \ldots, \tau_i$ for both policies. This flow of context is visualized by **solid arrows** $\rightarrow$. The **exploit policy** $\pi_{\text{exploit}}$ takes a context of episodes, and produces a single episode of exploitation. The return of this exploit episode is then used to train both policies, with the feedback to the explore policy visualized by the **dotted green arrows** $\cdots\!\!\rightarrow$. **B.** After the two policies are trained, different combinations of them are evaluated to find the combination that maximizes total reward. Each combination involves first exploring for $k$ episodes, and then repeatedly exploiting for the remaining $n - k$ episodes. **C.** The best combination is then used at inference time: exploring for a fixed number of episodes on new environments, and then exploiting for the remaining episodes.

return $G(\tau) = \sum_{t=0}^{h} \gamma^t R(s_t, a_t)$. Standard RL generally aims to learn a stochastic policy $\pi : \Omega \to A$ that maximizes the expected episode return $\mathbb{E}\left[G(\tau)\right]$.

Unlike standard RL, **meta-RL** trains an agent to: a) perform well in a distribution $\mathcal{D}$ of environments (e.g., a collection of mazes) and b) dynamically tailor itself to new environments (e.g. memorize a new maze over successive episodes). The agent completes meta-rollouts, each a sequence of episodes $\tau_1, \ldots, \tau_n$ on a new environment $E \sim \mathcal{D}$, with each episode within a meta-rollout beginning from a newly sampled start-state. The agent can remember information from earlier episodes in the same meta-rollout (e.g., using a sequence model such as a transformer [13]), allowing it to adapt its behavior based on that information. By training (via weight updates) on large batches of meta-rollouts, the agent then learns to leverage its memory mechanism, enabling sample-efficient memory-based adaptation and high performance on new environments sampled from $\mathcal{D}$.

Meta-RL methods can be split into two approaches that each solve a different problem, with specific algorithms designed and used for each approach [14][1]. **Cumulative-reward meta-RL** trains to maximize cumulative reward $\sum_{i=1}^{n} G(\tau_i)$. Examples include RL[2] [8, 9], VariBAD [7] and HyperX [10]. **Final-episode-reward meta-RL** aims to instead optimize solely for final episode reward $G(\tau_n)$. Examples include DREAM [11] and MetaCURE [15]. In this paper we compare and analyze First-Explore in a cumulative-reward setting. However, for the sake of completeness, we discuss how First-Explore relates to final-episode-reward meta-RL methods in Appendix I.

## 3 Related Work

Current cumulative-reward meta-RL methods train a single policy to maximize the cumulative reward of the whole episode sequence. However, optimizing directly for cumulative reward can prevent effective learning even on simple domains (e.g., bandits), as Section 1 describes and Section 6 demonstrates.

---

[1][14] use different terminology, calling cumulative-reward meta-RL "zero-shot meta-RL", and final-episode-reward meta-RL "few-shot meta-RL."

RL$^2$ [8, 9] is one of the first (cumulative-reward) meta-RL methods. It uses an RNN to provide across-episode memory, and standard RL algorithms to train the agent. RL$^2$ has the advantage of simplicity. By training standard RL algorithms with the capacity for across-episode memory, the agent may learn to optimally balance exploration and exploitation across successive episodes. However, training dynamics can hinder achieving optimal performance (e.g., due to the requirement of sacrificing immediate reward (Section 1 and Section 6) or because the reward signal is too sparse [10]).

Subsequently, various cumulative-reward meta-RL works have been produced. VariBAD [7] outperforms RL$^2$ in certain domains. VariBAD achieves this improvement via splitting training into producing a posterior belief of the current task (task inference), and a task-posterior conditioned policy. HyperX [10] improves exploration during policy training by adding an incentive to visit novel states. This exploration incentive is gradually decreased throughout training, starting high and ending at zero. The issue is that the cumulative-reward term can still *actively discourage* exploration, as described in Section 1. Therefore, HyperX primarily tackles the issue of sparse exploration rewards, rather than overcoming the challenge of domains that actively disincentivize exploration.

As Section 6 shows, when good early exploration requires forgoing immediate rewards, all these methods can fail to learn good behavior (e.g., converging immediately to a policy of never exploring). This failure arises not from reward sparsity (the problem HyperX tackles) or a lack of sophisticated posterior-belief-conditioned optimization (VariBAD's focus), but because the reward dynamics actively train the agent to *avoid* exploration. As such, First-Explore can achieve substantial total reward in domains in which RL$^2$, VariBAD, and HyperX all perform poorly.

AdA [12] demonstrates that meta-RL scales to complex domains and that training meta-RL on a curriculum of tailored learning challenges can produce agents capable of human-like adaption on complex held-out tasks. However, AdA may struggle to forgo reward, as the authors note that their agent always maximized current episode return (rather than the total reward of the episode sequence). This behavior suggests AdA's performance might depend on training and testing on environments that do not require such sacrificial exploration. Unfortunately, investigating AdA's dynamics is outside the scope of this paper.

## 4 First-Explore

First-Explore is a general framework for meta-RL that can be implemented in various ways. The framework trains two distinct policies, one policy to explore and one policy to exploit. Individually neither policy can achieve high cumulative reward, but *after* weight-update training, the policies are combined to produce an inference policy that does so. Figure 1 visualizes this process. By not *directly* training a single policy to maximize the total reward of the whole episode sequence, First-Explore avoids the failure mode of earlier approaches.

The **explore policy** $\pi_{\text{explore}}$ performs successive episodes. During each episode, it has access to a context containing all previous actions, observations, and rewards within the current exploration sequence. In contrast, the **exploit policy** $\pi_{\text{exploit}}$ is trained to take the context $c$ from the explore policy that has explored for a number of episodes between 1 and $n$, and to then run one further episode of exploitation. During this episode, the exploit policy has access to the actions, observations, and rewards from the current exploitation episode, as well as from the preceding exploration episodes (Figure 1).

The explore and exploit policies are incentivized differently. The **exploit policy** is incentivized to produce high-return episodes (based on the provided context of previous explorations). The **explore policy** is instead incentivized to produce episodes that each, *when added to the current context*, result in subsequent high-return exploit-policy episodes. Training the exploit policy requires context from the explore policy, and training the explore policy requires the returns of subsequent exploits. This is efficiently achieved by interleaving the policies as depicted in Figure 1-A, where each rollout from the explore policy is followed by a rollout from the exploit policy.

After the two policies are trained, First-Explore searches for a combination of them that maximizes the expected cumulative reward on newly sampled environments. Each candidate combination, first explores (via $\pi_{\text{explore}}$) for a set number of episodes $k$, and then exploits (via $\pi_{\text{exploit}}$) for the remaining $n - k$ episodes. These combination policies are evaluated on independent environments sampled from the target distribution, with the combination that maximizes mean cumulative reward being

chosen (see Appendix H for an example). This process of selecting $k$ does not involve retraining the policies, and thus it is comparatively computationally cheap. As such, $k$ should not be thought of as a hyperparameter, as unlike hyperparameters, the majority of the training compute expenditure is on policy weight updates that are performed *before* $k$ is chosen.

The resulting combination policy – first exploring for $k$ episodes and then exploiting for $n - k$ episodes – is then used as the inference policy $\pi_{\text{inference}}$ at test time, as shown in Equation 1.

$$\pi_{\text{inference}} = \begin{cases} \pi_{\text{explore}}, & \text{if } i \leq k, \text{ where } i \text{ is the episode number.} \\ \pi_{\text{exploit}}, & \text{otherwise} \end{cases} \tag{1}$$

## 5  Experimental Setup

First-Explore is implemented with a GPT-2-style causal transformer architecture [13], chosen for its strong temporal sequence modeling capabilities. To simplify the design, the explore and exploit policies share parameters, differing only in their final linear-layer head.

A novel sequence-modeling approach improved training stability in initial experiments, and was thus used for all First-Explore results. The method is described here and in the pseudocode provided in Appendix B. While this training method outperformed others in preliminary experiments, we believe the First-Explore meta-RL framework will work with alternate training approaches, such as PPO [16], Q-learning [6] and other RL algorithms.

### 5.1  Training Method

The **exploit policy** $\pi_{\text{exploit}}$ is trained to generate episodes that match or surpass the highest return achieved in prior episodes within the meta-rollout sequence. The **explore policy** $\pi_{\text{explore}}$ is trained to produce episodes that are followed by the exploit policy achieving higher episode returns than those seen so far. These training incentives implement the First-Explore framework: the exploit policy maximizes immediate episode returns, while the explore policy generates episodes that increase *subsequent* exploit-episode returns.

Training involves periodically updating the rollout policies $(\pi_{\text{explore}}, \pi_{\text{exploit}})$ with successor versions. These successor policies $(\phi_{\text{explore}}, \phi_{\text{exploit}})$ are trained to model the actions taken in the good rollout-policy episodes, defined as 1. good exploit episodes meet or surpass previous exploit returns in the meta-rollout sequence, and 2. good explore episodes are followed by the exploit policy achieving higher episode returns than those seen so far. We term these good exploit episodes 'maximal,' and the good explore episodes 'informative[2]'. Since the first exploit episode in each meta-rollout has no previous episodes for comparison, a baseline reward $b$ initializes the list of prior returns. This value is set as a domain-specific hyperparameter (but could easily be set automatically via heuristics).

At the start of training, both rollout policies are initialized with random weights. They are then copied to produce the initial successor policies.

Training is structured into iterated epochs. Each epoch, the current exploit and explore policies $(\pi_{\text{explore}}, \pi_{\text{exploit}})$ produce batched meta-rollouts. In each batch, the exploit episodes $\tau_{\text{exploit}} \sim \pi_{\text{exploit}}$ that are *maximal* and the explore episodes $\tau_{\text{explore}} \sim \pi_{\text{explore}}$ that are *informative* are recorded. For these criteria-satisfying episodes, a cross-entropy loss is calculated between the successor policies $(\phi_{\text{explore}}, \phi_{\text{exploit}})$ and the action distributions in the associated *maximal* or *informative* episodes. The successor policy weights are then updated, using gradient descent on the loss. In this way the successor policies learn to emulate the conditioned rollout policies (Equation 2). Finally, every $T$ epochs, the rollout policies $\pi_{\text{explore}}, \pi_{\text{exploit}}$ are updated to match the successor policies $\phi_{\text{explore}}, \phi_{\text{exploit}}$, ensuring continuous improvement. This hyperparameter $T$ manages the trade-off between preserving behavioral diversity and accelerating learning iteration.

$$\phi_{\text{explore}} \approx \pi_{\text{explore}} \mid \text{`informative episodes'}$$
$$\phi_{\text{exploit}} \approx \pi_{\text{exploit}} \mid \text{`maximal episodes'} \tag{2}$$

---

[2]As in all RL, improvement is a noisy process, (e.g., an episode labeled 'informative' might not be genuinely informative, due to random noise in the exploit reward). However, in expectation the labeling is correct.

During training, actions are sampled from their predicted probability distributions $a \sim \pi_{\text{exploit}}$ or $a \sim \pi_{\text{explore}}$. In contrast, during inference, actions are selected greedily.

## 6 Results

Table 1: Mean cumulative reward $\pm$ standard deviation of First-Explore compared against control algorithms in hard-to-explore domains, with random action (picking actions uniformly at random at each timestep) added for additional reference. In each domain, First-Explore significantly outperforms meta-RL controls. The bandit domain compares to two non-meta-RL baselines, marked †.

| Bandits with One Fixed Arm | | Dark Treasure Rooms | | Ray Maze | |
|---|---|---|---|---|---|
| First-Explore | $\mathbf{127.7 \pm 2.0}$ | First-Explore | $\mathbf{2.0 \pm 0.3}$ | First-Explore | $\mathbf{0.47 \pm 0.01}$ |
| RL$^2$ | $56.1 \pm 12.2$ | RL$^2$ | $0.2 \pm 0.1$ | RL$^2$ | $0 \pm 0$ |
| UCB-1$^\dagger$ | $116.8 \pm 0.5$ | VariBAD | $0.2 \pm 0.1$ | VariBAD | $0 \pm 0$ |
| TS$^\dagger$ | $123.3 \pm 2.5$ | HyperX | $-0.2 \pm 0.2$ | HyperX | $0.07 \pm 0.07$ |
| Random Action | $5.2 \pm 0.2$ | Random Action | $-5.5 \pm 0.1$ | Random Action | $-0.23 \pm 0.1$ |

When obtaining maximum cumulative reward requires forgoing early-episode reward, cumulative-reward meta-RL algorithms fail to learn optimal behavior regardless of how long they are trained (Appendix F), as they become stuck on a local optimum of not exploring well. Even simple environments such as bandits flummox them. Three varied domains empirically demonstrate this dynamic. On all three a) the meta-RL controls perform poorly, and b) First-Explore significantly outperforms the controls. Further, two of the domains have modified versions that do not require forgoing immediate rewards, and this change causes significant control policy improvement[3].

When forgoing immediate reward is required, First-Explore achieves 2x more total reward than the meta-RL controls in the first domain, 10x in the second, and 6x in the last (Table 1). These significant reward differentials reflect substantive behavioral differences, e.g., First-Explore exploring well (at the cost of reward) and then exploiting vs. the meta-RL methods converging to a policy of minimal exploration. First-Explore outperforming the other methods is also statistically significant, with $p$-values less than $10^{-5}$ for each comparison, as calculated by two-tailed Mann-Whitney $U$ tests (MWU).

**Domain 1: Bandits with One Fixed Arm:** The first domain is a multi-arm bandit problem designed to require forgoing immediate reward, where pulling the first bandit arm is immediately rewarding while also having no exploratory value. Pulling the first-arm yields a guaranteed reward of $\mu_1$, unlike the other arms, which – averaging across environments within the domain – yield expected reward 0. We consider two cases: $\mu_1 = 0.5$ (the deceptive case) and $\mu_1 = 0$ (the non-deceptive case).

For both values, the greatest cumulative reward can be reliably achieved by *first exploring* to find the highest reward arm (with reward greater than $\mu_1$) *and then exploiting* by repeatedly sampling it. However, $\mu_1 = 0.5$ creates the deceptive challenge described in Section 1, where *before reliable exploitation* has been learnt, exploration (i.e., not sampling arm 1 and obtaining its guaranteed reward) leads to a lower total reward. See Appendix C.1 for further details on this domain.

**Bandit Results (Figure 2):** First-Explore is evaluated against 3 baselines, with oracle performance (grey) and **random action selection (black)** plotted for additional reference.

**Deceptive Case:** The worst-performing control, RL$^2$ **[8, 9] (fuchsia)** is stymied by the deceptive domain dynamic. Four of five RL$^2$ training runs (overlapping bold fuchsia) learn to only sample the guaranteed reward, achieving a constant 0.5 reward each pull (Figure 2-B1), and exactly 50 reward after 100 pulls (Figure 2-A1). The remaining run (faint fuchsia) does better but still poorly. The best returns are achieved by methods that balance exploration (finding the most rewarding arm), and exploitation (pulling the best arm found so far). **First-Explore (green)** and the non-meta-RL bandit

---

[3]Surprisingly, First-Explore outperforms the controls even on the non-deceptive domains, $p < 10^{-5}$ (MWU). While promising grounds for future work, this paper only claims that First-Explore outperforms other methods in domains that require forgoing immediate reward.

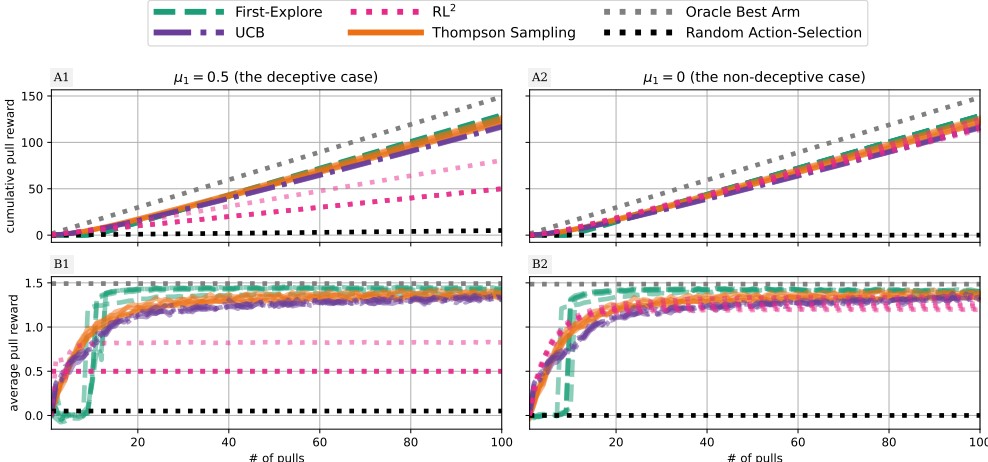

Figure 2: Mean performance (averaged across sampled bandits) of algorithms for deceptive (left) and non-deceptive (right) versions of the bandit domain. Each method trained 5 independent times, and each such run is plotted individually, so as to faithfully represent the variance between runs (e.g., that multiple of the bandit-domain $RL^2$ training runs achieve exactly the same reward). Appendix C provides alternative plots with mean reward $\pm$ standard deviation. The top figures plot the cumulative reward against the number of arm pulls, while the bottom figures illustrate the reward dynamics by plotting the individual pull rewards against the same. When the domain is deceptive, the cumulative-reward meta-RL method, $RL^2$ (fuchsia), performs extremely poorly, despite the deceptive domain giving strictly *higher* rewards than the non-deceptive version. In contrast, First-Explore (green) impressively outperforms UCB (purple) and Thompson Sampling (orange) despite them being specialized bandit algorithms, in both the deceptive and non-deceptive settings, with $p < 10^{-5}$.

controls, **Thompson Sampling [17] (orange)** and **UCB [8] (purple)**, both exhibit this behavior. Interestingly, First-Explore achieves the balance differently from the bandit algorithms, with the average pull rewards transitioning from close to zero, to close to optimal sharply around 10 pulls (Figure 2-B1). Impressively, First-Explore achieves greater reward than even the specialized bandit algorithms despite being applicable to more general domains ($p < 10^{-5}$, as calculated by MWU). HyperX and VariBAD are not included as baselines because their performance was overly poor even on non-deceptive versions of this domain[4]. **Non-Deceptive Case:** When the domain is not deceptive (i.e., when exploration does not require forgoing immediate reward), $RL^2$ achieves high total-reward (Figure 2-A2), and samples arms similarly to the bandit algorithms (Figure 2-B2). All other algorithms perform well regardless of deception. This dynamic validates that the need to forgo immediate reward is what stymies cumulative-reward meta-RL performance.

Illustrating that SOTA cumulative-reward meta-RL can fail on such simple domains (i.e., where standard-RL can easily solve each individual environment) is a key contribution of this paper.

**Domain 2: Dark Treasure Rooms:** The second domain is a grid world environment, where the agent cannot see its surroundings. In each environment, there are multiple randomly positioned reward locations. When each location is encountered, the agent consumes it for that episode, receiving a reward $\sim U[\rho, 2]$. The agent receives only its current coordinate as an observation, and to explore the agent must move blindly into unobserved grid locations that it has not visited before.

We consider two values of $\rho$: $\rho = -4$ (the deceptive case) and $\rho = 0$ (the non-deceptive case). For both values, high total reward is achieved by *first exploring* (visiting unseen grid locations to find positive reward locations) and *then __consistently__ exploiting* (revisiting discovered positive rewards locations each episode). However, when $\rho = -4$ this process is challenging, as only $\frac{1}{3}$ of the reward locations are positive, and the expected reward of a positive location is only $1$. These

---

[4]Despite hyperparameter tuning, good performance on either distribution (deceptive and non-deceptive) could not be achieved. Though bandits seem simple, the distribution of environment arm means is large and continuous. This may cause problems for VariBAD and HyperX, which were designed for and evaluated on small discrete task spaces, e.g. a goal being in one of 25 locations.

expectations create the challenge described in Section 1, where one must consistently exploit (by revisiting discovered positive reward objects more than three times) for any initial exploration to be worthwhile. See Appendix C.2 for an explanation of these properties, along with further domain details. In contrast, when $\rho = 0$, visiting unseen locations provides positive expected reward causing immediate rewards to incentivize (rather than deter) initial exploration.

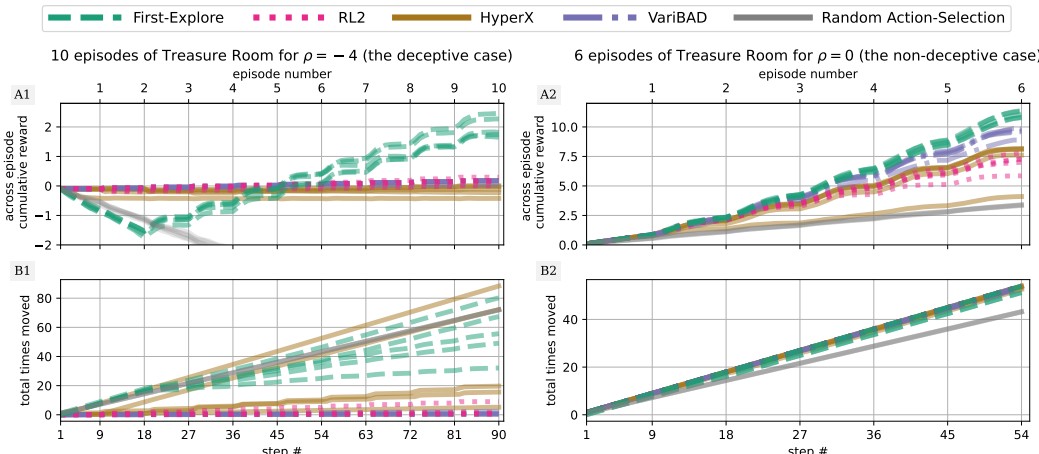

Figure 3: Mean performance (averaged across sampled treasure rooms) of algorithms for deceptive (left) and non-deceptive (right) versions of the Dark Treasure Room domain. Each method trained 5 independent times, and each such run is plotted individually. The top figures plot the cumulative reward obtained against step and episode number, while the bottom figures provide a proxy for exploration by plotting the number of times agents move against the same. When the domain is deceptive, the cumulative-reward meta-RL methods, $RL^2$ (fuchsia), HyperX (brown), and VariBAD (purple) achieve low total-reward, as the policies learnt to minimize exploration. In contrast, First-Explore (green) performs well on both the deceptive and non-deceptive domains.

**Dark Treasure Room Results (Figure 3):** First-Explore is evaluated against three meta-RL controls, with **random action selection (grey)** plotted for additional reference. **Deceptive Case:** The meta-RL algorithms $RL^2$ **(fuchsia)** and **VariBAD (pale lilac)** all achieve close to zero reward (Figure 3-A1), and rarely move (Figure 3-B1). **HyperX** explores more but minimally, as reflected by the modest negative first episode return[5]. However, the method fails to then exploit, and thus under-performs the other methods, despite this exploration.

By first exploring and then exploiting, **First-Explore (green)** achieves 10x more total reward in the ten-episode setting than the meta-RL controls. Impressively, despite the high cost of exploration (with a single episode of exploration taking more than three subsequent exploitations to yield positive expected reward), First-Explore is sufficiently skilled that its policy of exploring twice and then exploiting achieves more total-reward than a hand-coded policy of *optimally exploring* in the first episode, and then *optimally exploiting* conditioned on this initial exploration[6] $(2.0 > 1.78)$. **Non-Deceptive Case:** the meta-RL controls achieve reasonable total-reward (Figure 3-A2) and move consistently (Figure 3-B2), demonstrating that it is the difficulty of forgoing immediate rewards that challenges SOTA methods. However, First-Explore still outperforms them, with $p < 10^{-5}$ (MWU).

**Domain 3: Ray Maze:** The final domain is significantly more complex than the previous ones, and demonstrates that First-Explore can scale beyond bandit and grid-world problems. The domain is composed of randomly generated mazes with 3 reward locations. The agent must parse a large number of lidar observations (see Figure 4) to navigate around walls, identify reward locations, and move onto them. Each episode, the first reward location visited gives a reward of either $-1$ or $+1$, each reward location having an independent $30\%$ chance of positive reward. The agent has three discrete actions: rotate right, rotate left, and move forward. One complexity is that (unlike the

---

[5]The amount of exploration in the *first episode* is equal to the number of unique locations visited which is exactly proportional to the average negative first-episode return. As such, while HyperX moves a lot more than VariBAD and $RL^2$, it mostly moves into locations previously visited.

[6]See Appendix C.2 for a mathematical derivation.

grid-world domain), actions do not commute (e.g., rotating left and then moving forward is different from moving forward and then rotating left). The agent can see the reward locations, but cannot tell by looking whether they give positive or negative reward. Optimal behavior requires *first exploring* (navigating to visit un-visited reward locations, and obtaining their negative expected reward), and *then reliably exploiting* (re-visiting a reward location if it gave positive reward). The need to sacrifice immediate reward (by visiting the negative reward locations) challenges the SOTA cumulative-reward baselines.

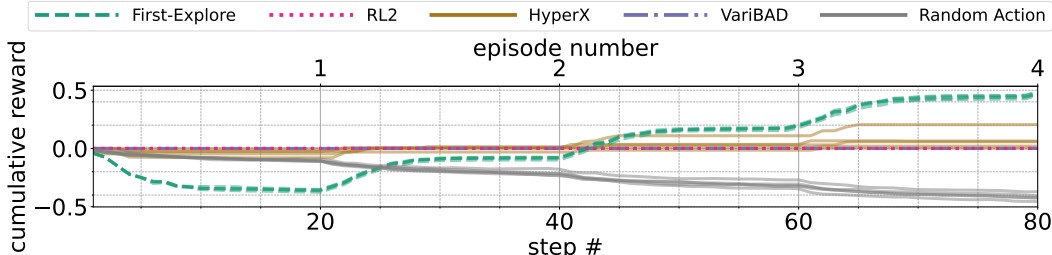

Figure 5: Mean performance averaged across 1000 Ray Mazes for five runs of each treatment. First-Explore strongly outperforms the SOTA meta-RL baselines on this complex environment, achieving a mean 0.47 reward, only slightly worse than the expected total-reward of behaving optimally, 0.64.

**Ray Maze Results (Figure 5):** On this more complex domain, First-Explore similarly achieves significantly more reward than the meta-RL controls, with all controls failing to perform well. Two of the three meta-RL controls, **RL$^2$ (fuchsia)** and **VariBAD (purple)**, learn to never move (and instead spin in-place). This behavior avoids encountering any reward location, and thus results in exactly 0 total reward in every evaluation of all five independent training runs. For the final meta-RL control, four of five **HyperX (brown)** runs move little (achieving near zero reward), with the remaining run achieving only modest cumulative reward.

**First-Explore (green)** significantly outperforms the controls, achieving 0.47 mean reward (Table 1), by exploring to find a reward location, remembering it and reliably exploiting by navigating to that location if it gave positive reward. Figure 5 reflects this process, illustrating that First-Explore's cumulative reward goes down substantially before then going up. For comparison, the optimum possible behavior of exploring and exploiting perfectly achieves at most 0.64 average reward[7].

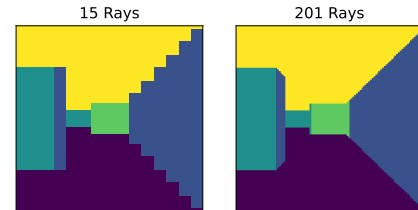

Figure 4: **Left:** Raw agent observations from a sampled ray maze converted to an image. The agent receives the wall distances and the wall types. Portraying this numerical data as an image, goal locations are **green**, and the two wall orientations are distinguished (**east-west teal**, and **north-south navy**). To aid the eye, the floor has been coloured in **dark purple**, and the **sky yellow**. Although the goal is visible, it could be a treasure (positive reward) or trap (negative reward). **Right:** The image produced with direct ray casting (large number of processed lidar measurements) rather than the 15 the agent receives.

## 7   Limitations and Future Work

As presented, First-Explore does not actively explore to enable future exploration, because the explore policy $\pi_{\text{explore}}$ only trains to increase the expected reward of the *subsequent exploit policy* episode. Unfortunately, a sequence of optimal myopic explorations is not necessarily an optimal exploration sequence (Appendix G). A potential solution is to reward exploration episodes based on a weighted sum of rewards from all subsequent exploitation episodes, analogous to summing discounted future rewards in standard RL.

Another limitation is that First-Explore could be unsafe in certain environments. The risk is that First-Explore's explore policy does not avoid negative rewards, and so penalizing unsafe action has minimal effect on the policy. For example, a chef robot attempting a new recipe in a physical home

---

[7]See Appendix C.3 for an upper bound on optimal performance.

might explore mistakes that endanger humans or damage the kitchen (despite negative rewards telling it not to). This concern only applies to a subset of environments, as many environments are safe, e.g., simulated ones. That said, addressing this concern is a valuable direction of future work. One potential solution is to add a safety penalty to the explore policy, i.e., train the explore policy to maximize *subsequent* exploit episode-returns while also avoiding safety risks. This proposed version of First-Explore could actually produce in-context adaption that is safer than standard RL training, as the meta-RL policies would have learnt a strong prior of avoiding potentially dangerous actions.

This paper makes no claims regarding First-Explore's meta-training efficiency, and future work may substantially improve meta-training time. Instead, this paper a) identifies a challenge that SOTA meta-RL algorithms fail on regardless of training time (the need to forgo immediate rewards), and b) provides a framework, First-Explore, that can solve these challenging domains.

A final problem is the challenge of long sequence modelling, with certain environments requiring a huge context and high compute (e.g., can one have a large enough context, and enough compute to allow First-Explore to generalize and act as a replacement for standard RL?). AdA [12] suggests such a project might be possible. Progress on efficient long-context sequence modelling [18, 19], research on RL transformer applications [20, 21], and Moore's Law all make this possibility more feasible.

Additionally, given that First-Explore learns a dedicated explore policy and a dedicated exploit policy, and both have been shown to work well (e.g., Figure 2), the method may be applicable to final-episode-reward meta-RL settings (Appendix I). Further, First-Explore always switches from exploration to exploitation after a fixed number of episodes. Future work could replace this fixed number with a learnt classifier that determines when to switch from exploration to exploitation, e.g., repeatedly exploring *until* sufficient information is obtained. However, despite these applications being highly exciting directions of future work, a proper investigation of either would require its own paper, including many different specialized controls and environments pertinent to the setting.

## 8    Discussion

Given that First-Explore uses RL algorithms to train the meta-RL policy, how might it solve hard-exploration problems that standard RL cannot, e.g. design a rocket for the first time? We believe that given a suitably advanced curriculum, and sufficient compute, First-Explore could learn powerful exploration heuristics (i.e., develop intrinsic motivations such as an analogue of curiosity) and that these heuristics would enable sample-efficient performance on hard sparse-reward problems. On a curriculum, initially First-Explore would explore randomly, and learn to exploit based on that random exploration. Once it has learnt rudimentary exploitation, the agent can learn rudimentary exploration. Then it would learn better exploitation and exploration, and so on, each time relying on there being 'goldilocks zone' tasks [22] that are not too hard and not too easy.

Further, while curricula can aid all of meta-RL, e.g., RL$^2$ and AdA, First-Explore can have a significant training advantage on certain problems (e.g., in the ten-episode Dark Treasure-Room, First-Explore achieves positive cumulative reward while the standard cumulative-reward meta-RL methods catastrophically fail). This advantage could potentially allow far greater compute efficiency, and allow training on otherwise infeasible curricula.

## 9    Conclusion

We have theorized and demonstrated that SOTA cumulative-reward meta-RL fail to train when exploration requires forgoing immediate reward. Even simple problems such as bandits (Figure 1) can stymie these methods. To overcome this challenge, we introduce a novel meta-RL framework, First-Explore, that learns two separate interleaved policies: one to first explore, another to then exploit. By combining the policies at inference time, First-Explore is able to explore effectively and achieve high cumulative reward on problems that hamstring SOTA methods.

Meta-RL shows the promise of finally fixing the main problem in RL – that it is extremely sample inefficient – even producing *human-level sample efficiency* [12]. However, the promise of this approach is limited, as we have shown, on a large set of important problems. We can only take advantage of this approach if we can harness the benefits of meta-RL even on such problems, and First-Explore enables us to do so, thus offering an important and exciting opportunity for the field.

## Acknowledgments and Disclosure of Funding

Resources used in preparing this research were provided, in part, by the Province of Ontario, the Government of Canada through CIFAR, and companies sponsoring the Vector institute `www.vectorinstitute.ai/#partners`. This work was further supported by an NSERC Discovery Grant, and a generous donation from Rafael Cosman.

Thanks to Michiel van de Panne, Mark Schmidt, Ken Stanley and Shimon Whiteson for discussions, and to Yuni Fuchioka, Ryan Fayyazi and Nick Ioannidis for feedback on the writing. We also thank Aaron Dharna, Cong Lu, Shengran Hu, and Jenny Zhang (sorted alphabetically) in our lab at the University of British Columbia for extensive discussions and feedback.

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

# Appendix

**Table of Contents**

# A  Replicability

To ensure full replicability, we are releasing the code used to train First-Explore and the controls, along with the environments trained on. We are also releasing the weights of a trained model for each domain. Each model contains both the explore and exploit policies as separate heads on the shared trunk. The code is available at `https://github.com/btnorman/First-Explore`.

# B  Training Pseudocode

```
def rollout(env, π, ψ, c_π, c_ψ):
    ### """perform a single episode
    # inputs: the environment (env),
    # the agent policy π, the successor policy ψ,
    # and the current policies' contexts c_π, c_ψ
    ### returns the episode return, temp_loss, and updated contexts"""
    temp_loss, r = 0, 0
    # n.b. temp_loss is only used if the episode meets a condition
    # see (*) in the conditional_action_loss function
    s = env.reset()  # state s
    for i in range(env.episode_length):
        # calculate action probabilities p_π, p_ψ for both policies
        # and update context
        p_π, c_π = π.action_probabilities(s, c_π)
        p_ψ, c_ψ = ψ.action_probabilities(s, c_ψ)
        a = sample_action(p_π)
        temp_loss += cross_entropy(a, p_π * p_ψ)   # hadamard product
        # * p_π ensures action diversity by weighting against likely actions
        s = env.step(s, a)
        r += s.reward
    return r, temp_loss, c_π, c_ψ

def conditional_action_loss(φ, θ, D, b):
    ### """calculates First-Explore loss for both exploring and exploiting
    # on domain D using the agent and successor parameters φ, θ
    ### and baseline reward b"""
    env = sample_env(D)
    π_explore, π_exploit = load_policies(θ)
    ψ_explore, ψ_exploit = load_policies(φ)
    c_π, c_ψ = set(), set()  # the agent and successor contexts
    loss, best_r = 0, b
    for i in range(D.episode_num):
        r_explore, l_explore, c_π, c_ψ = rollout(env, π_explore, ψ_explore, c_π, c_ψ)
        r_exploit, l_exploit, _, _ = rollout(env, π_exploit, ψ_exploit, c_π, c_ψ)
        # ^exploit context not kept
        # (*) accumulate loss if:
        if r_exploit >= best_r:  # exploit episode is 'informative'
            loss += l_exploit
        if r_exploit > best_r:   # explore episode is 'maximal'
            loss += l_explore
            best_r = r_exploit
    return loss
```

Algorithm 1: Example First-Explore Cross-Entropy Loss.

```python
def train(epoch_num, T, D, b):
    """example First-Explore training (ignoring batchsize)
    runs the meta-rollouts, accumulating a loss
    this loss is then auto-differentiated"""
    T_counter = 0
    φ = θ = init_params()
    for i in range(epoch_num):
        # θ is the agent behavior parameters
        # φ is the successor parameter (learning an improved agent policy)
        Δφ = ∂/∂φ (conditional_action_loss(φ, θ, D, b))
        φ -= step_size * Δφ
        # Update θ every T epochs
        T_counter += 1
        if T_counter == T:
            θ = φ
            T_counter = 0
    return θ
```

Algorithm 2: Example of Training First-Explore using the Cross-Entropy loss and Auto-Differentiation.

# C   Detailed Domains:

## C.1   Bandits with One Fixed Arm

**Domain Description:** Each bandit has ten arms, and in the environment the agent acts by selecting an arm to pull $a \in [1-10]$. The first arm always yields the reward $\mu_1$, while the other arms yield their environment specific arm mean $v_{a \in [2-10]} \sim \mathcal{N}(0,1)$ plus a normally distributed noise term $\mathcal{N}(0, \frac{1}{2})$, added to make the environments more challenging. The arm mean is fixed once the environment is sampled, but the noise term is resampled each arm pull. The objective is to maximize the expected reward obtained on a newly sampled bandit over 100 pulls.

**Additional Treatments:** Two specialized bandit algorithms were evaluated on this domain. The bandit algorithms: UCB-1 estimates an upper confidence bound and selects the arm that maximizes it, see Appendix J for details. Thompson Sampling (TS) [17] samples arm means from the posterior distribution, and chooses the arm with the best sampled mean.

See Figure 6 for a version of Figure 2 that plots mean $\pm$ standard deviation instead of each individual run.

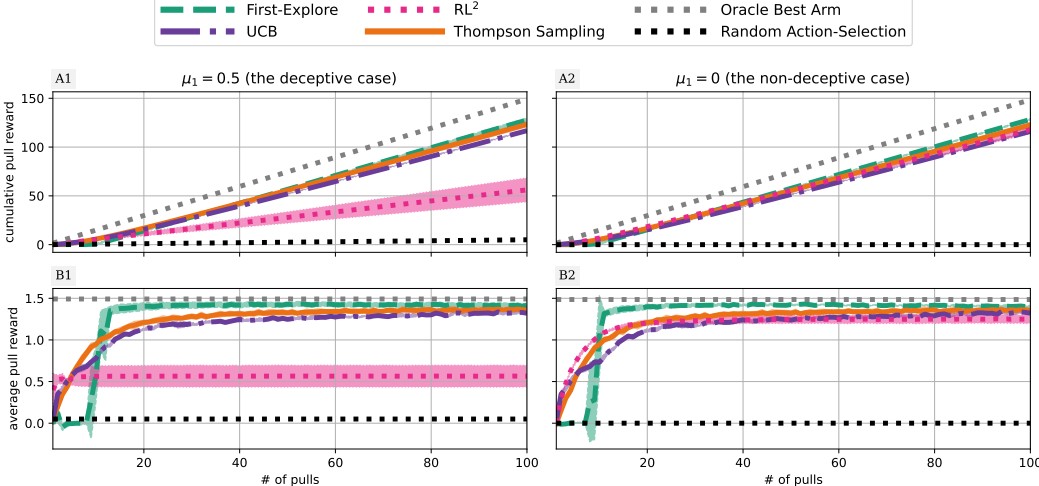

Figure 6: Alternative Bandits-with-One-Fixed-Arm Plots with Mean $\pm$ Standard Deviation.

## C.2 Dark Treasure-Rooms

**Problem Description:** Dark Treasure-Rooms (inspired by the Darkroom in [20]) are $9 \times 9$ grids full of treasures and traps. The agent starts in the middle of the grid, navigates (up, left, down, or right) to find treasure, and cannot see its surroundings. The agent can take five actions: four for cardinal movement, and one to not move. Only its current $(x, y)$ coordinates are observed. Each environment has $8$ objects (treasures or traps), and when the agent encounters a treasure or trap it consumes/activates it, and receives an associated reward (positive or negative). The treasure and trap values $v_i$ are uniformly distributed in the range $\rho$ to 2, $v_i \sim U[\rho, 2]$, with $\rho$ being the maximum trap penalty. The locations of the objects are sampled uniformly, with overlapping objects having their rewards/penalties summed. Each episode has 9 steps, and the objective is to maximize the expected cumulative episode returns on a newly sampled Dark Treasure-Room over multiple episodes.

See Figure 7 for a version of Figure 3 that plots mean $\pm$ standard deviation instead of each individual training run.

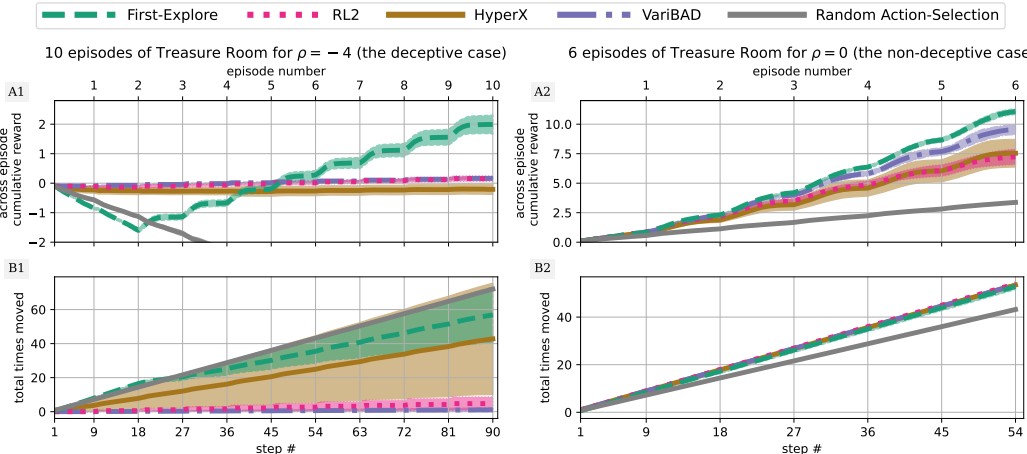

Figure 7: Alternative Dark-Treasure-Rooms Plots with Mean $\pm$ Standard Deviation.

### C.2.1 The $\rho$ Domain Dynamics

How difficult the Dark-Treasure room domain is strongly depends on the value of $\rho$. The dynamics are as follows:

- The fraction of positive rewards equals $\frac{2}{2-\rho}$.
- The average value of a positive reward is always $\mathbb{E}(U[0, 2]) = 1$ (assuming $\rho < 0$).
- The average value of a negative reward is $\mathbb{E}(U[\rho, 0]) = \frac{\rho}{2}$.
- The expected reward of visiting random reward location (treasure or trap) is $\mathbb{E}(U[\rho, 2]) = \frac{2+\rho}{2}$.

Using these, we can calculate the expected reward of visiting a random reward location and then returning to it $n$ times if and only if the location gave a positive reward. This expectation is equal to the expected reward of the first visit plus $n$ times the chance of the visited reward location giving positive reward multiplied by the expected positive reward (Equation 3).

$$\mathbb{E}(U[\rho, 2]) + n \frac{2}{2-\rho} \mathbb{E}(U[0, 2]) = \frac{\rho+2}{2} + n \frac{2}{2-\rho} \tag{3}$$

We can then calculate the number of revisits required for exploration (visiting a random reward location for the first time) to be worthwhile (i.e., lead to positive expected reward). Equation 4 illustrates this calculation, for values of $\rho < -2$. This number of revisits corresponds to how reliable the exploitation policy must be for exploration to yield higher cumulative reward. For example, when

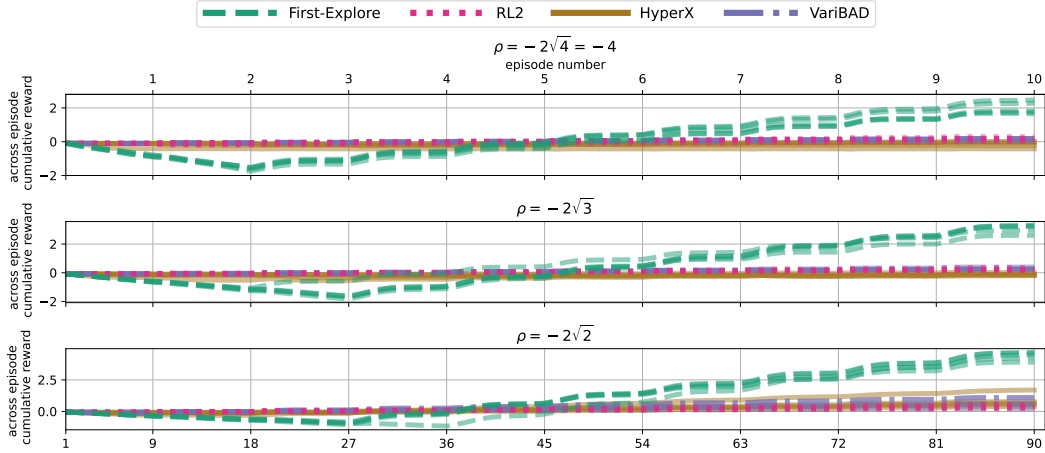

Figure 8: Behavior of First-Explore and the meta-RL controls on the 10 Horizon Dark Treasure Room for various values of $\rho$.

$\rho = -4$ on the ten horizon setting, the agent needs to exploit correctly more than $\frac{1}{3}$ of the time, as (after exploring) it has at most nine episodes to exploit, and must exploit correctly more than three times. If exploitation reliability is lower than this value, then exploration will be actively discouraged.

$$\frac{2+\rho}{2} + n\frac{2}{2-\rho} > 0 \tag{4}$$

$$4 - \rho^2 + 4n > 0 \tag{5}$$

$$n > \frac{\rho^2 - 4}{4} \tag{6}$$

When $\rho = -4$, $n > \frac{(-4)^2-4}{4} = 3$. Thus, $\rho = -4$ corresponds to exploration (visiting a new reward location) requiring more than three potential revisits to be worthwhile (have positive exploration). As (Section 1) describes, this dynamic challenges cumulative-reward meta-RL methods, as exploration thus requires reliable exploitation to be worthwhile.

This value of $\rho = -4$ was chosen as it a) requires a reasonably high number of revisits, and b) is an integer number that corresponds exactly to exploration being worthwhile only when the exploitation policy on average performs more than an integer number of revisits.

In contrast, as Section 6 describes, when $\rho = 0$, exploration is directly incentivized (and no revisits are required for exploration to increase the expected total reward).

For completeness, we also calculate the dynamics that occur when $\rho = -2\sqrt{2}$ and when $\rho = -2\sqrt{3}$, which corresponds to requiring more than 1 and more than 2 revisits respectively[8] (Figure 8 and Table 2). Even when the domain is made significantly less challenging (i.e., for more positive values of $\rho$), First-Explore signficantly outperforms the controls.

---

[8]As found by solving Equation 3, $\rho = -2\sqrt{x}$ corresponds to exploration (visiting a random treasure location) requiring more than $x - 1$ revisits to yield increased total reward.

Table 2: Mean cumulative reward $\pm$ standard deviation on 10 Episode Dark Treasure Rooms, for different values of $\rho$. $\rho = -2\sqrt{x}$ corresponds to exploration (visiting a random treasure location) requiring more than $x - 1$ revisits. Even when the domain is made significantly less challenging (i.e., for more-positive values of $\rho$), First-Explore signficantly outperforms the controls.

|  | $\rho = -2\sqrt{2}$ | $\rho = -2\sqrt{3}$ | $\rho = -2\sqrt{4}$ |
|---|---|---|---|
| First-Explore | $\mathbf{4.4 \pm 0.3}$ | $\mathbf{3.1 \pm 0.3}$ | $\mathbf{2.0 \pm 0.3}$ |
| RL$^2$ | $0.4 \pm 0.1$ | $0.2 \pm 0.1$ | $0.2 \pm 0.1$ |
| VariBAD | $0.7 \pm 0.3$ | $0.3 \pm 0.1$ | $0.2 \pm 0.1$ |
| HyperX | $0.8 \pm 0.5$ | $-0.1 \pm 0.1$ | $-0.2 \pm 0.2$ |

#### C.2.2 One Optimal Exploit episode, and then Subsequent Optimal Myopic Exploitation

In this domain, perfect exploration in the first episode corresponds to visiting 9 unique coordinates, as each coordinate has an equal chance of being a reward location (a treasure or a trap). This process discovers $9 * \frac{8}{81} = \frac{8}{9}$ reward locations on average, as there are 8 reward locations spread over 81 coordinates, and the agent visits 9 of the coordinates.

Assuming the agent then optimally myopically exploits (i.e., only maximizing current-episode reward and not attempting to explore more), the total reward is at most $\frac{8}{9}$ times the expression in Equation 3 (the expected number of reward locations multiplied by the expected value of perfectly exploiting each individually).

Equation 7 calculates this expectation for $\rho = -4$ and $n = 9$ (the values of the deceptive Dark Treasure Room domain in Section 6). Note, $n$ is the number of *subsequent* exploitations, after visiting a new reward location, and so the $n = 9$ calculation corresponds to the ten episode setting.

$$\mathbb{E}(\text{explore optimally then optimally myopically exploit}) \leq \frac{8}{9}\left(\frac{\rho + 2}{2} + n\frac{2}{2 - \rho}\right) \quad (7)$$

$$\leq \frac{8}{9}(-1 + 9\frac{1}{3}) \quad (8)$$

$$\leq 1.7\overline{7} \approx 1.78 \quad (9)$$

This calculation is an upper bound, as it assumes the agent can effectively teleport to discovered positive reward locations. In reality, the agent may not be able to revisit some discovered positive-reward locations without moving into discovered negative-reward locations.

## C.3 Ray Maze

**Problem Description**: Ray Maze challenges the algorithms with a more complex domain. The agent must navigate a randomly generated maze of impassible walls (the maze layout is different for each sampled Ray Maze) to find one of three goal locations. Each goal location is either a trap or a reward, with a 0.3 probability of the goal being a treasure, and a 0.7 probability of it being a trap. Treasures give reward 1, while traps give penalty $-1$. In this domain, the agent can only receive one goal reward per episode, as triggering the first prevents others activating. To perceive the maze, the agent receives 15 lidar observations (see Figure 4). Each lidar observation reports the distance to the nearest wall along an angle (relative to the agent's orientation). It further tells the agent whether the lidar ray hit a goal location, as well as the wall orientation (east-west or north-south). However, the lidar measurements do not show whether a goal is a trap or a treasure. The agent has 3 actions, turn left, go forward, and turn right, with rotation turning the agent's field of view.

Ray Maze is a challenging domain for several reasons. It has a high-dimensional observation space (15 separate lidar measurements), complex action dynamics (with actions not commuting, e.g. turn left then move forward is different from move forward then turn left) and a randomly generated maze that interacts with both movement and observation. Furthermore, similarly to earlier environments, the agent must learn from experience, and risk the traps in early episodes, so as to exploit and consistently find treasure in later ones. Because of how frequent traps are, the agent can only obtain positive cumulative reward by a) searching for treasures in early episodes (at the cost of expected reward, as the average value of a goal is negative) and b) reliably exploiting in later episodes (navigate to identified treasures, while avoiding potential traps).

See Figure 9 for a version of Figure 5 that plots mean $\pm$ standard deviation instead of each individual training run.

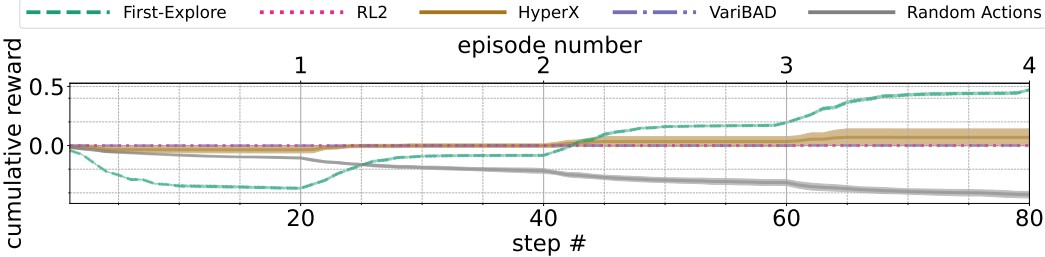

Figure 9: Alternative Ray-Maze Plot with Mean $\pm$ Standard Deviation.

### C.3.1 Optimal Total-Reward

Despite being a complicated environment, Ray Maze has simple reward dynamics.

- Each goal location has an independent 0.3 chance of yielding +1 reward, otherwise it gives -1 reward.

- All goal locations are independent, and the agent can only receive one goal reward each episode

Thus, **perfect exploration** within an episode involves always visiting a goal location (potentially finding a location that gives positive reward), and **perfect exploitation** involves always visiting a positive reward location if one is known, while avoiding all negative and unknown goal locations.

Assuming the agent always revisits positive goal locations, the expected reward of visiting an new goal location with $n$ episodes remaining is $-0.4 + 0.3n$ (the expected value of the first visit, plus the expected value of revisiting positive rewards $n$ times).

- In the first episode the expected value of visiting a *new* goal location is $-0.4 + 0.3 \times 3 = 0.5$.

- In the second episode, the expected value is $-0.4 + 0.3 \times 2 = 0.2$.

- In the third episode, the expected value is $-0.4 + 0.3 \times 1 = -0.1$, and so not worthwhile.

Thus the optimal policy is to visit a goal location in the first episode (and then repeatedly revisit it if there was a positive reward), and then if the first goal location was negative (70% chance), visit a goal location in the second episode.

This policy achieves $0.5 + 0.7 * 0.2 = 0.64$ expected total reward. This value is an upper bound, as it is possible that the agent cannot always find a goal location.

# D  Tabulated Results:

Table 3: Bandits with One Fixed Arm Results. The bandit domain compares to two non-meta-RL baselines, marked †.

|  | $\mu_1 = 0.5$ (the deceptive setting) | | $\mu_1 = 0$ (the non-deceptive setting) | |
|---|---|---|---|---|
|  | mean $\pm$ std dev | median | mean $\pm$ std dev | median |
| First-Explore | $\mathbf{127.7 \pm 2.0}$ | $\mathbf{128.4}$ | $\mathbf{128.4 \pm 0.7}$ | $\mathbf{128.6}$ |
| RL$^2$ | $56.1 \pm 12.2$ | $50$ | $117.9 \pm 3.9$ | $117.0$ |
| UCB-1$^\dagger$ | $116.8 \pm 0.5$ | $116.7$ | $116.1 \pm 0.6$ | $115.8$ |
| TS$^\dagger$ | $123.3 \pm 2.5$ | $122.5$ | $122.7 \pm 2.4$ | $121.9$ |

Table 4: Dark Treasure-Room Results

|  | $\rho = -4$ (the deceptive setting) | | $\rho = 0$ (the non-deceptive setting) | |
|---|---|---|---|---|
|  | mean $\pm$ std dev | median | mean $\pm$ std dev | median |
| First-Explore | $\mathbf{2.0 \pm 0.3}$ | $\mathbf{1.8}$ | $\mathbf{11.1 \pm 0.2}$ | $\mathbf{11.0}$ |
| RL$^2$ | $0.2 \pm 0.1$ | $0.2$ | $7.2 \pm 0.7$ | $7.4$ |
| VariBAD | $0.2 \pm 0.1$ | $0.2$ | $9.6 \pm 0.4$ | $9.8$ |
| HyperX | $-0.2 \pm 0.2$ | $-0.2$ | $7.5 \pm 1.2$ | $8.0$ |

Table 5: Ray Maze Results

|  | mean $\pm$ std dev | median |
|---|---|---|
| First-Explore | $\mathbf{0.5 \pm 0.01}$ | $\mathbf{0.47}$ |
| RL$^2$ | $0 \pm 0$ | $0$ |
| VariBAD | $0 \pm 0$ | $0$ |
| HyperX | $0.07 \pm 0.07$ | $0.06$ |

# E  Compute Usage

Each training run commanded a single GPU, specifically a Nvidia T4, and up to 8 cpu cores. Table 6 gives the approximate walltime of each run.

Table 6: Compute Usage Per Training Run. Many of the meta-RL controls converged early, and did not improve with longer periods of training time (see Appendix F). Domains where this occurred are marked †. To save compute, these runs were not trained as long as First-Explore.

| Run | Runtime |
| --- | --- |
| Meta-RL Deceptive Bandits First-Explore | 20 hours |
| Meta-RL Deceptive Bandits RL$^2$ | 40 hours (extended, see below) |
| Dark Treasure-Room First-Explore | 50 hours |
| Dark Treasure-Room HyperX & VariBAD & RL$^2$ | 10 hours† |
| Ray Maze First-Explore | 75 hours |
| Ray Maze HyperX, VariBAD, and RL$^2$ | 10 hours† |

Notably, in the Dark Treasure-Room for $\rho = -4$, VariBAD and RL$^2$ rapidly converges to a policy of staying still, and while HyperX seems to slowly improve rewards, the reward increase is an artifact caused by the HyperX having an exploration incentive that is gradually attenuated to zero. This attenuation creates a slow convergence from negative reward (due to moving into traps and not exploiting) to a higher near zero reward (obtained by mostly staying still). Because the attenuation is designed to occur throughout the entire run, scaling the run length merely scales how long HyperX takes to converge to close to zero reward.

Due to a desire to not waste compute on converged policies, once this behavior was verified, the control runs on this setting were limited to 10 hours. In contrast, First-Explore was run for longer, as it continued to improve with additional training. This is a fair comparison, as due to the controls having converged, increasing the control run training time would not yield better policies, as Figure 10 demonstrates. To this end, as RL$^2$ on the bandit environment still showed improvement after 10 hours, those runs were also extended.

Total compute used for the experiments would then be around 1100 hours (5 runs for each treatment). However, there was also hyperparameter search, e.g., for the RL$^2$ bandit parameters. As such, total compute may be over 2000 GPU hours. Furthermore, there were many preliminary experiments to iterate on the First-Explore architecture as well as to research and identify cumulative-meta-RL deception.

# F  Poor Performance Regardless of Train Time:

The issue is not that the methods are slow to converge, and that with more training they could perform well. As Section 1 describes, these controls achieve low cumulative reward regardless of how long the methods are trained. Figure 10 demonstrates this phenomenon on the deceptive treasure room.

**Figure 10-Top:** RL$^2$, VariBAD, and HyperX average cumulative reward plotted against training time. RL$^2$ (yellow) and VariBAD (gray) converge to zero reward almost immediately. This transition corresponds to the policies learning to stay still. HyperX (teal) reward increases (toward zero) throughout meta-training. However, this increase in reward comes not from HyperX learning an increasingly sophisticated policy, but instead is the result of the HyperX algorithm's meta-training exploration bonus being linearly reduced from the start to the end of meta-training. Thus, once that bonus is near zero, HyperX also learns to stay still.

**Figure 10-Bottom:** HyperX with different training lengths (specified by number of episode steps). When HyperX is run for ten times as long (orange) or ten times less long (blue) than the default training time (light blue) the same behavior is observed (of slow convergence to (slightly below) zero reward). This behavior demonstrates the improvement in reward comes from the HyperX algorithm reducing the exploration incentive during the meta-training. It also implies that changing the length of training runs (including running for much longer) would not change the final performance results.

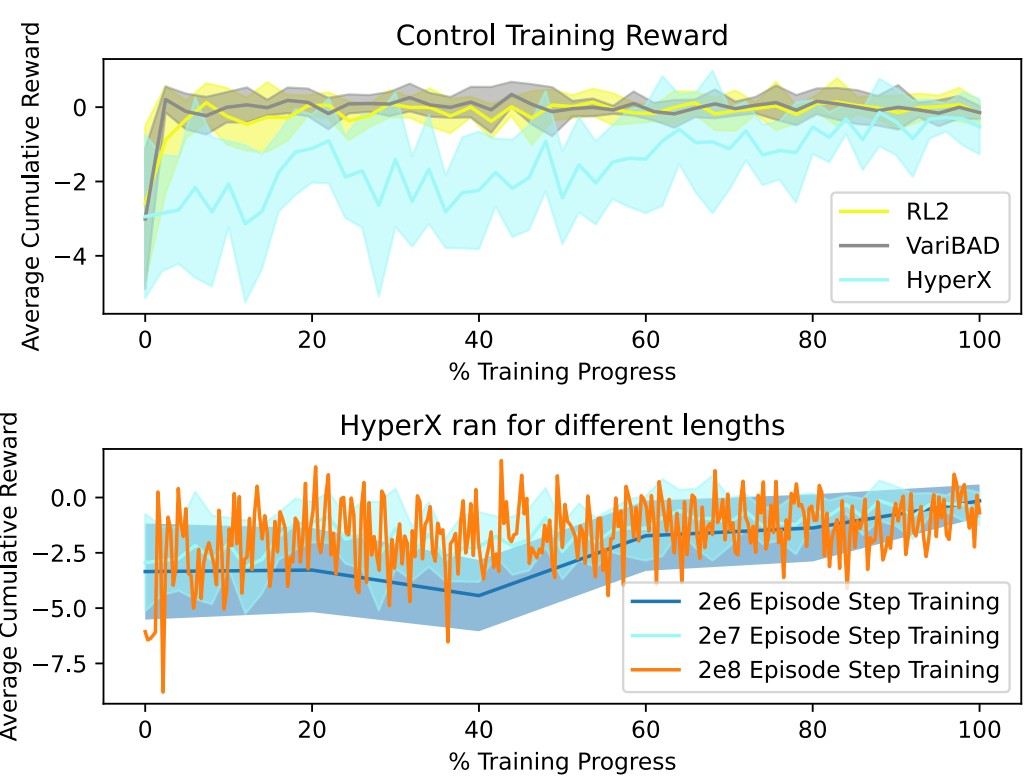

Figure 10: Regardless of training time, the meta-RL controls perform poorly.

# G   Myopic Exploration

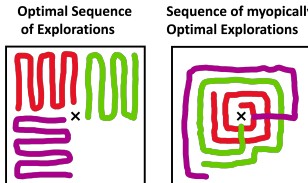

Figure 11: Well-planned sequential exploration can sometimes significantly outperform a sequence of optimal myopic explorations. For example, consider exploring a plain over four days, where each day one must explore by walking from the plain's center. **Sequence of Optimal Myopic Explorations** one 'optimal' way of exploring is to perform a spiral from the center (e.g., the red spiral on the right). This strategy achieves the optimal amount of exploration on day 1 as one never retraces one's steps. However, if one does a spiral on day 1 then on day 2, one must retread old ground - wasting time otherwise spent exploring new locations. Each day bee-lining to unseen areas and then spiralling from there is also optimal for that day, however it increases the amount of retreading tomorrow. **Optimal Sequence of Explorations:** another optimal way of exploring on day 1 is to explore a quadrant, visualized in red on the left. Again, as one does not backtrack, this strategy is optimal on day 1. However, unlike the spiral strategy, this strategy is also part of an optimal sequence of four explorations, as one can explore a new quadrant each of the four days, without ever retreading the same ground.

# H   K-Selection Phase

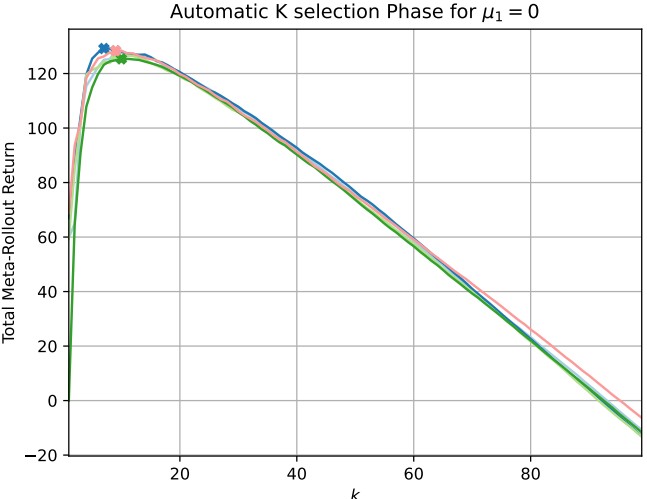

Figure 12: Demonstration of First-Explore's $k$-selection phase, for the bandit distribution, with $\mu_1 = 0$. Five separate First-Explore runs are plotted. The training runs select different values of $k$ (due to the relative strengths of each runs explore and exploit policy), with the associated selected $k$ corresponding to the peak of each curve (marked by a cross).

# I Final-Episode-Reward Meta-RL

Methods such as MetaCURE [15], EPI [23] and CCM [24] learn an exploration policy that aims to extract maximum environment information (independent of whether such information informs good exploitation). These approaches discard grounding exploration in (maximizing) future reward. Not grounding exploration in future exploitation reward means that the policy may learn (via weight updates) to spend meta-rollouts acquiring irrelevant information (e.g., the exact penalty of bad actions). This distraction potentially prevents optimal exploration from ever being learnt.

E-RL$^2$ [25] modifies RL$^2$ to ignore the first-$k$ episode rewards. This modification enables pure exploration (that is not dissuaded by negative rewards). However, E-RL$^2$ introduces an across-episode value assignment problem: identifying which exploration episodes enabled good subsequent exploitation. This problem potentially limits training sample efficiency. Further, the exploratory episodes number $k$ is set as a hyperparameter and constant across all tasks (both at training and at inference), preventing efficient combination with a curriculum that contains different difficulty tasks (as hard tasks may need significantly more exploration episodes than easy ones). Finally, hard coding $k$ limits the flexibility and usefulness of E-RL$^2$ because one cannot explore until a satisfactory policy quality is reached, preventing meta-RL in-context adaptation from off-the-shelf replacing standard RL.

DREAM [11] also separately optimizes exploration and exploitation policies (and grounds exploration in exploitation), but has four complex, manually designed, interacting components and a reliance on knowing unique problem IDs during meta-training. This complexity enables increased sample-efficiency by avoiding the chicken and egg problem of simultaneously learning explore and exploit policies. Unlike E-RL$^2$, because a part of DREAM's machinery must learn to produce the right information per problem based on the (unique, random) problem ID only, it is unable to generalize or handle never-seen-before challenges during meta-training, raising questions about its scalability and generality. For example, DREAM may potentially be difficult to apply to problems where each training environment is unique (e.g., for environments with continuous variables, or samples from otherwise vast search spaces). It may also struggle when each environment is a hard-exploration challenge, as it may be difficult for the model to explore enough to learn which information is required to solve the problem. We believe curricula are necessary to solve such environments. However, because DREAM cannot generalize during meta-training (as described above), it cannot take advantage of a curriculum to build an exploration skill set to tackle harder and harder exploration challenges.

Applying First-Explore to a final-episode-reward meta-RL setting is a promising direction of future work, as First-Explore i) learns grounded exploration ii) can explore until sufficient information is obtained ( rather than having a fixed number of explorations), and iii) does not rely on privileged information (problem IDS), allowing generalization during meta-training.

# J Evaluation Details

Evaluation (sampling the multiple evaluation environments and performing iterated rollouts) was with a single GPU. For the **Bandit Results**, each of the First-Explore evaluations sampled $128 \times 100$ bandit environments, indepedent from those trained on. For the 5 RL$^2$ bandit evaluations batch size was reduced to $2,000$ (due to taking longer to evaluate). Since there is no meta-RL training variance for UCB1 and TS, five independent evaluations were done, each with an independently sampled $10,000$ bandits.

**UCB**: UCB was implemented according to the description in [8], with $c = 1$. Namely, each pull, UCB picks the arm that maximizes $\text{ucb}_i(t) = \hat{\mu}_i(t-1) + \sqrt{\frac{2 \log t}{T_i(t-1)}}$, where $\hat{\mu}_i(t-1)$ is the estimated mean reward of the $i$th arm, $T_i(t-1)$ is the number of times the $i$th arm has been pulled

For the **Dark Treasure-Room** and the **Ray Maze** domains, all policies were evaluated on a batch of $1,000$ environments sampled independently from those trained on.

# K  Training Details

## K.1  Controls:

The official VariBAD [7] (VariBAD and RL$^2$) and HyperX [10] (HyperX) codebase ran the meta-RL controls. **Dark Treasure-Room** trained with the default hyperparameters of the coded bases gridworld environments. These were found to perform well, with variations tried not yielding improvement. To provide a strong control on the **Bandits with One Fixed Arm** problem, the controls were advantaged by having individual hyperparameter gridsearches for each $\mu_1$ value (unlike First-Explore). See the SI attached configuration file for the exact hyperparameters. Both of these codebases are licensed under a MIT license.

## K.2  First-Explore:

The architecture for both domains is a GPT-2 transformer architecture [13] specifically the Jax framework [26] implementation provided by Hugging Face [27], with the code being modified so that token embeddings could be passed rather than token IDs. The different hyperparameters for the two domains are given in Table 7. The code being provided with a Modified MIT License (allowing free use with attribution).

For all domains each token embedding is the sum of a linear embedding of an action, a linear embedding of the observations that followed that action, a linear embedding of the reward that followed that action, a positional encoding of the current timestep, and a positional encoding of the episode number. See the provided code for details. For the dark treasure-room environments a reset token was added between episodes that contained the initial observations of the environment, and a unique action embedding corresponding to a non-action. The bandit domain had no such reset token.

Table 7: Model Hyperparameters

| Hyperparameter | Bandit | Dark |
|---|---|---|
| Hidden Size | 128 | 128 |
| Number of Heads | 4 | 4 |
| Number of Layers | 3 | 4 |

For training we use AdamW [28] with a piece-wise linear warm up schedule that interpolates linearly from an initial rate of $0$ to the full learning rate in the first 10% training steps, and then interpolates linearly back to zero in the remaining 90% of training steps. Table 8 gives the optimization hyperparameters.

Table 8: Optimization Hyperparameters

| Hyperparameter | Value |
|---|---|
| Batch Size | 128 |
| Optimizer | Adam |
| Weight Decay | 1e-4 |
| Learning Rate | 3e-4 |

Hyperparameters were chosen based on a relatively modest amount of preliminary experimentation. Finally, for efficiency, all episode rollouts and training was done on GPU using the Jax framework [26].

For evaluation, we then sample by taking the argmax over actions, and do not add the $\epsilon$-noise.

Table 9: Training Rollout Hyperparameters

| Hyperparameter | Bandit | Darkroom | Ray Maze |
|---|---|---|---|
| Exploit Sampling Temperature | 1 | 1 | 1 |
| Explore Sampling Temperature | 1 | 1 | 1 |
| Policy Update Frequency | every training update | every $10,000$ | every $5,000$ |
| $\epsilon$ chance of random action selection | 0.05 | 0 | 0 |
| Baseline Reward | 0 | 0 | 0 |
| Training Updates | 200,000 | 1,000,000 | 1,000,000 |

# L   Dark Treasure-Room Visualizations

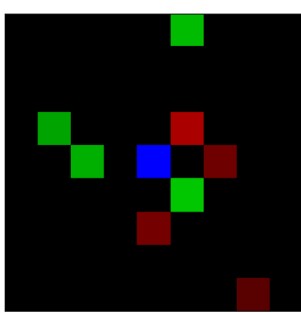

Figure 13: A visualization of the dark treasure-room. The agent's position is visualized by the blue square, positive rewards are in green, and negative rewards are in red, with the magnitude of reward being visualized by the colour intensity. When the agent enters a reward location it consumes the reward, and for that timestep is visualized as having the additive mixture of the two colours.

Here are example iterated First-Explore rollouts of the two trained policies, $\pi_{\text{explore}}$, $\pi_{\text{exploit}}$, visualized for a single sampled darkroom.

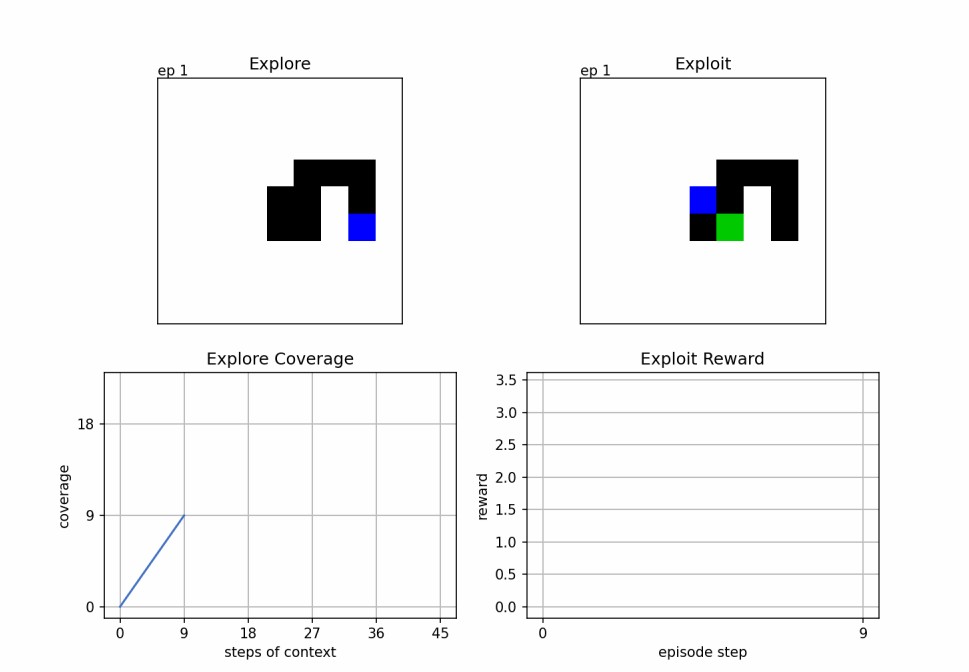

Figure 14: The first (First-Explore) explore episode. **Top left** visualizes the last step of a First-Explore explore episode, with the locations that are not in the cumulative context being coloured white, as the agent is blind to them (having no observations or memory of those locations). This figure plots the end of the first exploration, and shows a reward has been found. **Bottom left** visualizes the coverage of the cumulative context by plotting the total number of unique locations visited by the exploration against the cumulative episode step count. In this explore, the agent never doubled back on itself, which is good as it is optimal to have as many unique locations visited as possible. **Top right** visualizes a step in a First-Explore exploit episode, with the locations that are in context visualized. The agent can effectively 'see' those locations in its memory. **Bottom right** plots the exploit reward against the exploit episode timestep. As this figure plots before the start of the exploit episode, the agent has yet to move and encounter rewards, but will have done so in the subsequent visualizations.

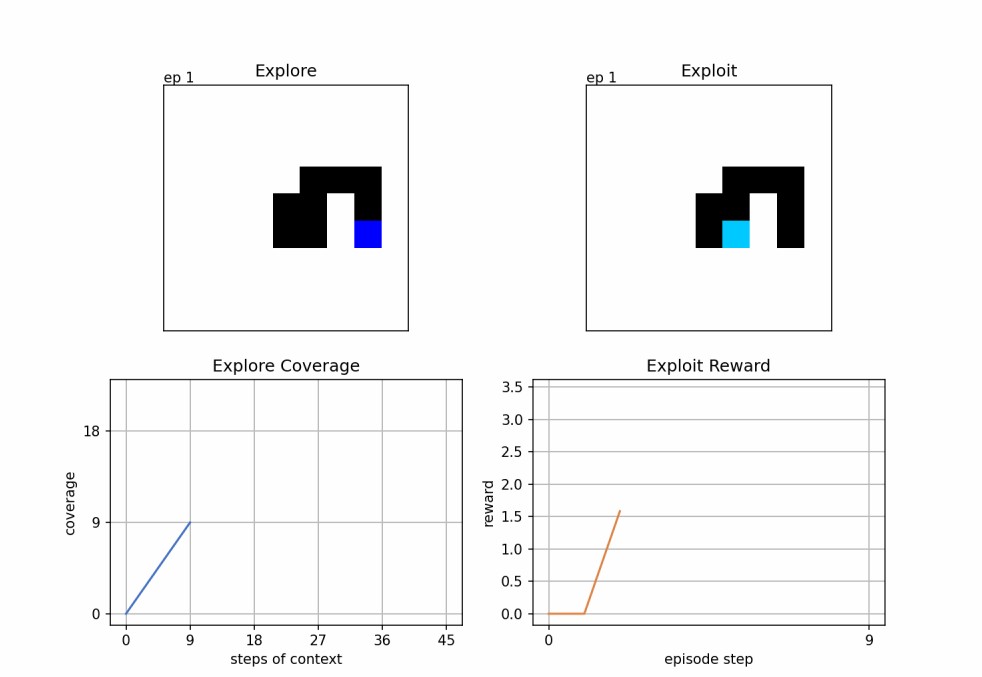

Figure 15: The first (First-Explore) exploit episode. This figure uses the same visualization design as Figure 14. **Left top and bottom** are the same as in Figure Figure 14, and of the explore context, not the current exploit episode. **Right top**, the agent (the light blue square) has found the reward in the first two steps. Consuming the reward is visualized by the agent colour and the reward colour being combined. **Right bottom**, the associated episode reward is shown.

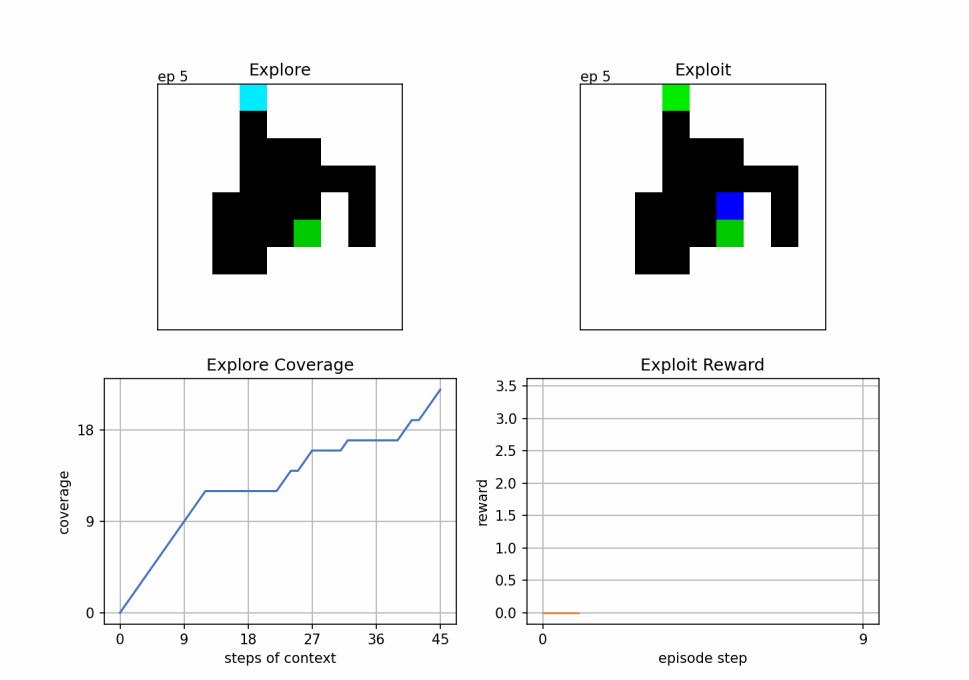

Figure 16: The fifth (First-Explore) explore episode. At the end of the 5th explore episode the agent has discovered a new positive reward at the top of the room, and can now 'see' it in memory. The new information presents an opportunity for the exploit policy to obtain both rewards, but it only has exactly enough time-steps in an episode to navigate to do so, and thus cannot make a mistake navigating.

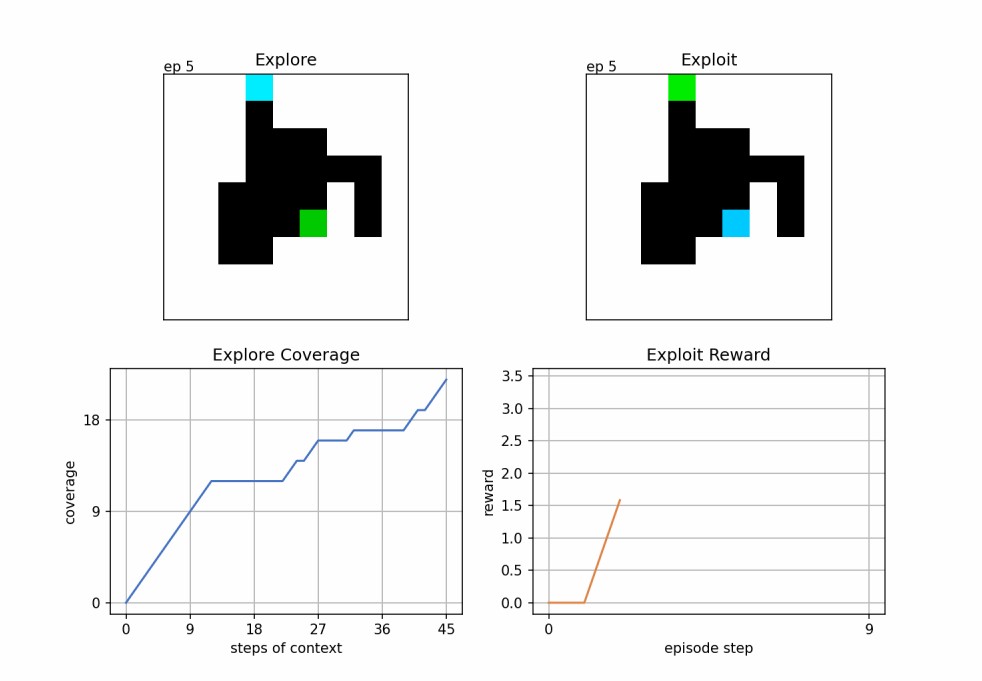

Figure 17: The first reward of the fifth (First-Explore) exploit episode. Two steps into the episode the agent (in consuming, light blue) has consumed the nearby reward.

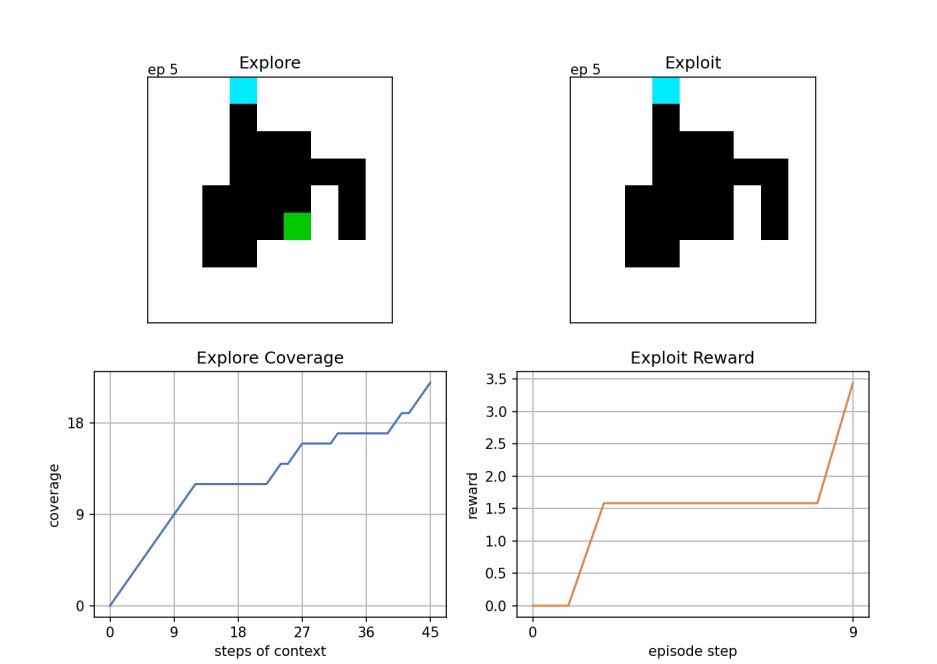

Figure 18: The end of the fifth (First-Explore) exploit episode. After consuming the nearby reward the agent has reached the newly discovered reward at the top of the room and consumed it. This success required making no mistakes and navigating first to the nearby reward then to the top one on the first try. This inference is possible because the quickest the agent can reach both rewards is exactly the length of the episode (9 steps). The navigation in this episode is an example of intelligent exploitation, as after the information reveal (the reward at the top) of a single episode the agent appropriately changes its policy based on the context and using the learnt environment prior (e.g., how to navigate), produces a highly structured behavior (navigating with no mistakes).

