# OpenReview forum: "First-Explore, then Exploit: Meta-Learning to Solve Hard Exploration-Exploitation Trade-Offs"
_NeurIPS.cc/2024/Conference — NeurIPS 2024 poster_

### Official Review · Reviewer_zDt1 · 2024-07-08

**Soundness:** 3
**Presentation:** 3
**Contribution:** 3
**Rating:** 6
**Confidence:** 3

**Summary:**

This paper proposes a method to mitigate the issue of deceptive rewards in Meta-RL, I.e., rewards that may impede further exploration leading to higher cumulative rewards. The authors propose to learn two policies: an exploration and exploitation policy that are conditioned on a common context. During policy training the reward from the exploitation policy is fed through the exploration policy via the context as feedback. In the next phase the amount of necessary exploration iterations are determined through the cumulative reward. When the optimal cutoff is found the final policy is the combination of the exploration and exploitation policy based on the optimal cutoff $k$. The authors test on a bandit problem and dark-treasure room environment, mitigating the issue of deceptive rewards compared to other known Meta-RL approaches.

**Strengths:**

* Meta-RL is a challenging and important problem in reinforcement learning. The problem of deceptive rewards in Meta-RL seems to be an inherent problem of Meta-RL worth solving.
* The idea of conditioning on different exploration contexts to determine the exploration cutoff is a simple but elegant idea.
* The proposed method is effective at mitigating the problem of deceptive rewards for the tested environments.
* The paper is generally well written and I understood the motivation behind tackling this problem.

**Weaknesses:**

* The choice of $\rho=4$ seems a bit arbitrary. I think $\rho$ should be ablated to see what the effect of different penalties is on the proposed method and other methods is.
* The environments chosen for the experiments seem very simplistic. It is difficult to assess whether the method would succeed when having to learn long-horizon and more complex policies. In this case it might become intractable to get feedback from the exploitation policy  to decide on the right amount of exploration steps.
* I think a good test would also have been environments such as MiniGrid or Minihack for example, which have also been used for hierarchical reinforcement learning.  The required sequence of actions is more complex than the proposed environments, while still being observationally simple.

**Questions:**

* How do you think your method scales to more complex environments? Would it be possible to maybe make a more complex dark-treasure-room?

* In the caption of Figure 4 I don’t understand the following statement:
	> By first exploring for two episodes using πexplore, and then exploiting 	using πexploit, First-Explore’s inference policy πinference achieves significant cumulative reward...”

* I thought this cutoff was determined automatically and also considering the environments are procedural why does the value of 2 work for all the configurations? Shouldn’t this value vary depending on where the different objects are spawned?

* Why does the performance of your method go down first in Figure 4 while this does not happen for the other methods?

**Limitations:**

I think the authors have properly addressed the limitation of their work.

---

> ### Author Rebuttal · Authors · 2024-08-07
>
> Thank you for reading, engaging with, and critiquing our work. We greatly appreciate the time you have spent, and your feedback has enabled us to improve the paper. We are delighted that you support sharing First-Explore with the NeurIPS community and consider it well-motivated and addressing “a problem worth solving.”
>
> We have addressed your concerns and substantially strengthened the paper. We hope you will consider our improvements and raise your score to further support its acceptance.
>
> # Weakness Response:
> > [The choice of $\rho$ seems arbitrary.]
>
> Setting $\rho=0$ ensures all objects have non-negative rewards, so exploring (visiting new locations) only increases rewards. This setting provides the simplest non-deceptive control, and demonstrates that the baselines (and First-Explore) can handle problems without local optima.
>
> Setting $\rho = -4$ ensures that repeated and consistent exploitation is necessary for exploration to be worthwhile, thus creating local optima. Evaluating in this domain examines First-Explore’s central claim: that these exploration dynamics are deceptive and challenging to existing cumulative-reward meta-RL methods, and that First-Explore overcomes this challenge.
>
> When $\rho = -4$, the expected reward of an object is -1 (as the rewards are distributed U[-4, 2]). Furthermore, the expected reward of an object with positive reward is only 1 (as they are distributed U[0, 2]), and only $\frac{1}{3}$ of objects have positive reward. This means multiple exploitations are required before exploration pays off. If an object is found while exploring, the expected reward is -1. However, optimal exploitation yields an expected $\frac{1}{3}$ reward (by avoiding traps and navigating to positive reward objects). Thus, it takes at least four exploitation episodes for exploration to be worthwhile, as $4 \times \frac{1}{3} - 1 = \frac{1}{3} > 0$.
>
> We will include an explanation of these choices in the paper.
>
>  > [how would behavior vary with $\rho$?]
>
> Between $\rho=-4$ and $\rho=0$ there will be a transition point between the two results shown in the paper. When $\rho=-4$, the baseline fails catastrophically while First-Explore performs well (see Figure 4 A1). When $\rho=0$, the baselines perform reasonably, although First-Explore still outperforms them (see Figure 4 B1). While exciting (and a promising direction of future work), First-Explore outperforming the baselines in the non-deceptive case is not central to the paper’s claims. Increasing $\rho$ from 0 would not qualitatively affect the results since the environment cannot become more non-deceptive. Decreasing $\rho$ cannot worsen baseline performance since they already fail, but it might eventually harm First-Explore due to the sparsity of rewarding objects. We will add text to the paper noting this, along with supporting plots.
>
> > [What about more complex environments?]
>
> We have added a more complex domain, Ray Maze (see the main response to all reviewers for details). This new domain is significantly more complex than the Dark Treasure Room. However, the same pattern of performance holds: First-Explore outperforms all controls, which fail catastrophically due to the deceptive nature of the domain.
>
> >[How about environments such as MiniGrid or Minihack for example, which have also been used for Hierarchical Reinforcement Learning?]
>
> Thank you for the suggestion. We do not consider First-Explore an example of Hierarchical Reinforcement Learning, which typically involves high-level and low-level policies (e.g., a policy setting subgoals and another achieving them). First-Explore operates by exploring first, then exploiting, without one policy setting subgoals for another. Therefore, MiniGrid and MiniHack would not test the main scientific hypotheses of this paper.
> # Question Response:
> > [Is a more complex dark treasure room possible?]
>
> Indeed it is, as we now show! Please see the main PDF, and our main response for details on the new Ray Maze domain.
> > Why does the value of 2 work for all the configurations? Shouldn’t this value vary[...]?
>
> There may be confusion about when First-Explore switches from exploring to exploiting. After training the explore and exploit policies, the number of exploration episodes $k$ is determined by evaluating on a new batch of environments. Given $k$, the inference policy explores for the first $k$ episodes (regardless of exploration performance) before switching to the exploit policy. Surprisingly, in deceptive domains, this simple approach outperforms all controls. We will clarify this in the text.
>
> What you describe (the exploit policy deciding on the right amount of exploration steps) is a promising direction of future work that we will add to Section 6.
>
> > Why does [First-Explore performance] go down first in Figure 4 [and not for] other methods?
>
> When $\rho = -4$, the environments are such that the expected reward of an unseen location (one not visited in a prior episode) is negative, and thus moving in the first episode leads to negative expected reward. Positive cumulative rewards can be achieved because, over multiple episodes, the agent can exploit any positive rewards it has found while avoiding negative ones. Unfortunately, the other methods fail to learn this strategy and instead learn to mostly stay still (so avoiding negative reward in the first two episodes, but also preventing the significant cumulative reward First-Explore achieves by the end).
>
> The reward goes down for 2 episodes (vs. another number), because $k$ was automatically determined to be 2 on this domain. This setting causes the resulting inference policy to explore for the first 2 episodes (and so seek unseen negative-expected-value locations) before then exploiting for the remaining 8 episodes.

---

> > ### Comment · Reviewer_zDt1 · 2024-08-12
> >
> > I appreciate the authors efforts in addressing my questions.
> >
> > - Regarding Minihack and Minigrid, I didn't think your method was Hierarchical Reinforcement Learning based, but these environments lend themselves to test your scenario as well and have been quite established in the literature. However, this is a minor point for me.
> >
> > - Overall, with the added experiments and explanations, I think this is a decent paper that addresses a gap when deceptive rewards have to be foregone Meta-RL. Overall, it seems to me this is a still unexplored issue in RL in general.
> >
> > I will raise my score to 6 and keep my confidence at 3

---

> > > ### Author Response · Authors · 2024-08-12
> > > **Thank You**
> > >
> > > We are sincerely grateful for your questions and the time you invested engaging with our work, and we are delighted that you value the additional experiments and explanations. We greatly appreciate your score increase, and hope that (in tandem with the other reviewers and the AC) it leads to this work being published and thus shared with the NeurIPS community.

---

### Official Review · Reviewer_Erp8 · 2024-07-12

**Soundness:** 3
**Presentation:** 2
**Contribution:** 2
**Rating:** 6
**Confidence:** 3

**Summary:**

This work proposes a novel Meta-RL framework called First-Explore to address the balance between exploration and exploitation. The method learns two distinct policies: one for exploration and one for exploitation. These two policies are then combined to form the final inference policy. The effectiveness of the method has been demonstrated in bandit tasks.

**Strengths:**

1. The problem of balancing exploration and exploitation in RL is highly relevant, and the proposed solution provides valuable insights.

2. The experimental results are comprehensive, demonstrating the effectiveness of the proposed method across bandit tasks.

**Weaknesses:**

1. The writing of the paper needs more careful organization. Figures 1 and 2 are not cited in the main text, which makes it difficult to understand their relevance. Additionally, the training process for the explore and exploit policies is not well explained in Section 4.

2. The inference policy appears to require exhaustive hyperparameter tuning (k), which limits the method's generalization and makes it challenging to apply to other tasks.

3. The compared baselines seem outdated. It would be beneficial to provide comparison results with more recent methods to better demonstrate the effectiveness of the proposed approach. Additionally, the experimental results are not significant, as the proposed method does not outperform other baselines by a substantial margin in most tasks.

**Questions:**

1. What is the practical objective of the exploration policy, and how can the informativeness of an episode be defined?

2. I am curious about the training efficiency of the proposed method. Is it more efficient than learning an individual policy for each task?

**Limitations:**

The authors have discussed the limitations.

---

> ### Author Rebuttal · Authors · 2024-08-07
>
> Thank you for reading, engaging with, and critiquing our work. We greatly appreciate the time you have spent doing so, and your feedback has enabled us to strengthen the paper. We are delighted that First-Explore tackles “a highly relevant” problem, that the “proposed solution provides valuable insights,” and that the experimental results are “comprehensive.”
>
> We have addressed your concerns and substantially improved the paper. Furthermore, we believe your current positive comments support a higher score. We hope you will consider our improvements and increase your score by 2 to 3 points. Without this score increase, the paper may not be selected for publication.
>
> # Weaknesses Responses
> > [Figures 1 and 2 are not referenced]
>
> The paper now references Figure 1 and Figure 2.
> > The training process for the explore and exploit policies is not well explained in Section 4.
>
> Did you mean Section 3.2? Section 4 discusses related work. To address this issue, we have added pseudocode detailing the training process (see main reviewer response for details) and improved the writing in Section 3.2.
> > The inference policy appears to require exhaustive hyperparameter tuning (k), which limits the method's generalization and makes it challenging to apply to other tasks.
>
> As mentioned in Section 3.2, lines 154-157, “k is not a hyperparameter as, unlike hyperparameters, all policy-weight updates are performed independently of k, precluding the need to train the explore and exploit policies multiple times (which is the majority of the training compute-expenditure).” As illustrated in Figure 2, after training, k is automatically chosen (with minimal compute) via a quick interval search on a validation set. It can thus be quickly updated if the test distribution shifts.
> > [The compared baselines seem outdated.]
>
> VariBAD and HyperX are state-of-the-art cumulative-reward meta-RL methods. This paper exclusively makes claims on cumulative-reward meta-RL behavior, as opposed to final-episode-reward meta-RL. Similarly, RL$^2$ is still frequently used for cutting edge research (e.g., AdA) and remains an important baseline due to its simplicity and power.
> > [the results are not significant]
>
> The results have a large effect size, are extremely statistically significant, and fully support the claims of the paper. In short, the results are highly significant in every sense.
>
> The paper’s central thesis is that in deceptive domains existing cumulative-reward meta-RL methods fail, and that First-Explore overcomes this challenge. In the deceptive version of all three domains – the bandits when $\mu_1=0.5$ (Figure 3 A1), the Dark Treasure Room when $\rho=-4$ (Figure 4 A1), and the Ray Maze (see main reviewer response) – the meta-RL controls achieve abysmal performance. In these deceptive domains, First-Explore achieves significantly higher rewards than the best performing meta-RL control: 128.36 vs. 56.12 (> 2x improvement), 1.99 vs. 0.16 (> 10x improvement), 0.46 vs. 0.06 (> 7x improvement) for the Bandit results, Dark Treasure Room results, and Ray Maze results respectively. This large effect size reflects a significant and meaningful difference in agent behavior; the controls have learnt to minimize movement and exploration (Figure 4 B1), while First-Explore explores initially at the cost of reward (Figure 3 B1 and Figure 4 A1). All the results are also extremely statistically significant ($p < 2 \times 10^{-5}$). We shall modify the paper to make the results’ significance clearer.
>
> # Question Responses
> > What is the practical objective of the exploration policy, and how can the informativeness of an episode be defined?
>
> Regarding the practical objective of the exploration policy and the definition of informativeness of an episode: From Section 3.2, “the explore policy π explore is trained to produce episodes that are followed by the exploit policy achieving higher episode returns than those seen so far. These episodes are termed ‘informative.’”
>
> Training the explore policy to produce these ‘informative’ episodes (ones that are followed by the exploit policy achieving higher episode returns than it would have otherwise) is detailed in the pseudocode now added to the paper (see the main response). We will also improve the writing throughout the paper to clarify these issues.
>
> > [Is First-Explore efficient?]
>
> Is it more efficient than standard-RL applied to each environment?
>
> As the introduction states, “Meta-RL can potentially [be more sample efficient than standard-RL by] expending a large amount of compute to train a single agent that can then rapidly adapt to new environments, even showcasing human-like sample efficiency.” Once the initial compute expenditure is made then the trained algorithm can be extremely efficient on new (in-distribution) tasks at inference time (i.e., when adapting to new tasks).
>
> Is learning two policies more efficient than learning a single policy?
>
> Learning two policies enables First-Explore to avoid catastrophic failure in the deceptive domains. In this sense, First-Explore is far more efficient than the controls, as they fail to learn good behavior regardless of how long they are trained (e.g., in the hostile Dark Treasure Room Domain, RL2 and VariBAD converge to not moving, and HyperX exhibits the same poor performance regardless of training time, see Figure 5 in Appendix C).

---

> > ### Comment · Reviewer_Erp8 · 2024-08-09
> > **Response to Author's Rebuttal**
> >
> > Thank you for the clarifications provided. However, I still have some confusion regarding the paper:
> >
> > **1. Training Process:**
> > - **Distinction between Policies:** I don't see the difference between $\pi_{explore}$ and $\pi_{exploit}$, as both seem to load from $\theta$. Could you please clarify this?
> > - **Loss Calculation:** Why is $l_{explore}$ added twice to the loss when $r\underline{\text{}}exploit$ is greater than or equal to $best\underline{\text{}}r$?
> > - **Role of Predictor:** What is the role of the predictor? Is it similar to the target policy?
> > - **Definition of "Informative":** Is "informative" defined as achieving a higher return? What if the agent receives negative rewards before reaching the objective but receives positive rewards when deviating from the objective? Does the concept of "informative" still work in this context?
> >
> > **2. Inference:**
> > I am unclear on how $k$ can be automatically chosen. Figure 2B states that "after the two policies are trained in the previous phase, the optimal number of explorations is estimated by selecting the number of explorations $k$ that leads to the highest mean cumulative-episode reward on a large batch of new (unseen) environments sampled from the target distribution." This suggests to me that $k$ is selected through tuning. If I have misunderstood something, please correct me.
> >
> > **3. Experimental Results:**
> > I am not an expert in meta-RL and may not be up-to-date with the latest research. If other reviewers do not consider it necessary to compare with newer methods, I am fine with that. However, Figures 3 and 4 could be improved by plotting the mean and standard deviation ranges. Multiple lines for one method make it somewhat difficult to follow the intended message. I can see that First-Explore outperforms other Meta-RL methods. If the authors can clarify my questions regarding the training and inference processes, I am willing to raise my score.

---

> ### Author Response · Authors · 2024-08-10
> **Question Answers Part 1**
>
> Thank you for your continued feedback. We are delighted you are open to raising your scores to support the paper's publication. We hope our responses below (in this and Part 2) fully address your concerns. Please feel free to reach out if you have further questions.
> # Your questions:
> > [How are $\pi_{explore}$ and $\pi_{exploit}$ different, as they both load from $\theta$?]
>
> $\theta$ is a container holding the parameters of both policies. The two policies, $\pi_{explore}$ and $\pi_{exploit}$, differ because, while they share some parameters, they each have unique parameters. In particular, our implementation shares all parameters in the same neural network except for the last layer, see lines 161-162 in Section 3.2.
>
> We will update the pseudocode to clarify that $\theta$ represents a combination of parameters for both $\pi_{explore}$ and $\pi_{exploit}$. A comment will also be added to the load_policies function stating that each policy is constructed using its relevant subset of $\theta$.
> > Why is l_explore added twice to the loss when r_exploit is greater than or equal to best_r?
>
> To clarify, l_explore and l_exploit are distinct terms in the loss function, and each is added only once under specific conditions related to r_exploit and best_r. Is it possible you mistook l_explore as l_exploit at some point?
>
> Quoting from the main rebuttal,
> ```python
> if r_exploit > best_r:
> 	loss += l_explore
> if r_exploit >= best_r:
> 	loss += l_exploit
> ```
> To clarify, l_explore is added to the loss function if the exploit return (r_exploit) exceeds the best exploit return observed so far. This addition encourages the explore policy to produce episodes that enhance the exploit policy's performance.
>
> Similarly, l_exploit is added if the exploit return meets or exceeds the best exploit episode return in the sequence. Here, the loss condition incentivizes the exploit policy to achieve high episode returns. Notably, both conditions check the return of the exploit policy, but add different terms to the overall loss. We will emphasise this detail in the pseudocode.
>
> > [Is $k$ chosen automatically?]
>
> After training, the optimal value of $k$ is determined by automatically evaluating each candidate $k$ within the interval $[1,n−1]$ where $n$ is the total number of episodes. To evaluate each of these values of $k$, the associated inference policy (explore $k$ times, then exploit $n-k$ times) is run on a batch of target-distribution environments, and the $k$ that achieves the highest mean cumulative reward is selected. This selected $k$ is then used for inference at test time.
>
> We will rephrase the quoted text to clarify the automatic nature of the $k$ selection process, eliminating any implication of manual tuning. We will also make sure to clarify that this automated selection of the value of $k$ happens after training, and is computationally extremely inexpensive vs. training.
>
> > [What is the role of the predictor?]
>
> In our implementation, the agent uses its policy parameters $\theta$ to perform rollouts. These parameters are only periodically updated. A second set of parameters, the predictor's parameters $\phi$ learn an improved policy and are updated every batch of environments. Every $T$ learning updates, the agent policy is set equal to the predictor policy. This setup increases behavioral diversity, and enhances training stability.
>
> We will expand the pseudocode and methods section to clarify the distinct roles of the agent and predictor parameters.

---

> > ### Author Response · Authors · 2024-08-10
> > **Question Answers Part 2**
> >
> > > Is "informative" defined as achieving a higher return?
> >
> > “Informative” is defined for the exploration policy (only). It means those exploration episodes that (when the *exploit* policy gets to condition on them) are followed by the exploit policy achieving a higher return (than it would have otherwise, i.e. the exploit policy gets a higher return than it gets without adding this new explore episode to its context, meaning there is valuable information in that explore episode, hence it being “informative”).
> >
> > > What if the agent receives negative rewards before reaching the objective but receives positive rewards when deviating from the objective? Does the concept of "informative" still work in this context?
> >
> > We are not sure what you are asking. Are you asking if the episodes labeled as “informative” are always truly informative?
> >
> > The process is noisy. On average, genuinely informative episodes (e.g., ones that do meaningfully inform the exploit policy) are most likely to be followed by an increase in reward and hence be labeled as “informative.” However, not all “informative” episodes will contain useful information. For example, the exploit policy might happen to achieve high reward (e.g., from the bandit noise term), despite not receiving new information from the explore policy, and the code would still label the explore-policy episode “informative,” and incentivize more such explorations.
> >
> > It should be noted that this type of problem applies to all of RL. For example, a value function can erroneously identify an action as being high-value due to noise in the reward signal. Despite this noise, the concept is still valid, and as our results demonstrate, there is sufficient signal for the explore policy to learn effectively.
> >
> > We will add text to Section 3 explaining this dynamic.
> >
> > > [Figures 3 and 4 should display the mean and standard deviation]
> >
> > Due to the non-normal distribution of our training run returns, representing variability with standard deviation could potentially mislead readers. For example, in the deceptive bandit domain, four out of five $RL^2$ runs consistently select the same arm, leading to a highly skewed distribution. To accurately represent the variability across runs, the individual training runs are plotted instead. However, for completeness, we will include alternative plots with mean and standard deviation ranges in the appendix, accompanied by an explanation of their limitations.

---

> ### Comment · Reviewer_Erp8 · 2024-08-10
> **One More Question**
>
> Thank you for your quick and clear response. I have one last question: Does First-Explore belong to the meta-learning framework?
>
> To my knowledge, the meta-learning framework operates on the principle of "learning to learn." For example, one learner acquires knowledge that helps another learner to perform a specific task. Typically, the relationship between these two learners is unidirectional. However, the two policies, $\pi_{explore}$ and $\pi_{exploit}$, in your approach seem to learn from each other, making their relationship bidirectional.
>
> Is my understanding incorrect?

---

> > ### Author Response · Authors · 2024-08-11
> > **Response**
> >
> > Thank you for your question. First-Explore is an example of meta-reinforcement learning (meta-RL), which is an instance of meta-learning. Similar to RL$^2$, First-Explore operates within the meta-RL framework (see the definition below) by using machine learning to develop an inference policy. Once produced, this policy then functions as a “reinforcement learning algorithm,” because, as stated in lines 120-122 of Section 2, the policy adapts to new environments by leveraging memory of previous episodes (and associated rewards) to enhance performance in successive episodes (e.g., learning to pull the bandit arm that yields the highest mean reward). We will modify the text to further emphasize this process of adaptation, and how it enables “learning to (reinforcement) learn.”
> >
> > The bidirectional relationship between $\pi_{explore}$ and $\pi_{exploit}$ (with each relying on the other to train) is a unique feature of our approach but does not affect its nature as meta-RL (and thus meta-learning).
> >
> > To clarify terms:
> >
> > Meta-learning is a term used broadly across various areas of machine learning, where there is some sense of “learning to learn.” For instance, Oreshkin et al. [2020] describes meta-learning as “*usually linked* to being able to (i) accumulate knowledge across tasks (i.e. transfer learning, multi-task learning) and (ii) quickly adapt the accumulated knowledge to the new task (task adaptation)”
> >
> > Meta-RL is a subset of meta-learning (corresponding to “learning to *reinforcement* learn”) and is more concretely defined. Beck et al. [2023] write, “meta-RL uses sample-inefficient ML to learn sample-efficient RL algorithms, or components thereof.” To elaborate, the “learning” is separated into two phases, a sample inefficient initial phase that produces an RL algorithm, and the RL algorithm (which can then “learn” by performing successive rollouts in environments). This structure is in First-Explore, as it trains the inference policy (by learning both $\pi_{explore}$ and $\pi_{exploit}$ and then combining them). The resulting inference policy then acts as a highly sample-efficient reinforcement learning algorithm (see above, along with the results in Section 5, and the main rebuttal).
> >
> > Oreshkin, B. N., Carpov, D., Chapados, N., & Bengio, Y. (2020). Meta-learning framework with applications to zero-shot time-series forecasting. arXiv. https://arxiv.org/abs/2002.02887
> >
> > Beck, J., Vuorio, R., Liu, E. Z., Xiong, Z., Zintgraf, L., Finn, C., & Whiteson, S. (2023). A Survey of Meta-Reinforcement Learning. arXiv. https://arxiv.org/abs/2301.08028

---

> ### Comment · Reviewer_Erp8 · 2024-08-12
>
> Thank you for addressing my questions. I appreciate the effort put into clarifying my concerns, and based on your responses, I am willing to raise my score to 'weak accept'.

---

> > ### Author Response · Authors · 2024-08-12
> > **Thank You**
> >
> > We sincerely value your questions and the time you invested engaging with our work. Your feedback led to revisions that significantly strengthened our paper. We are delighted that our responses have clarified your concerns and appreciate your support for sharing our work with the NeurIPS community.

---

### Official Review · Reviewer_Vncg · 2024-07-13

**Soundness:** 2
**Presentation:** 2
**Contribution:** 2
**Rating:** 5
**Confidence:** 4

**Summary:**

This paper addresses the challenge in meta-reinforcement learning, where agents struggle to perform intelligent exploration across episodes, often failing to avoid repetitive exploration of the same locations. The authors propose a method called First-Explore, which learns two distinct policies: one focused on exploration and another on exploitation. By separating these policies, First-Explore aims to overcome the limitations of existing cumulative-reward meta-RL methods, which maximize cumulative rewards rather than maximizing the rewards of the final episode and, hence, not exploring adequately. The proposed method improves performance in environments like deceiving bandits and dark treasure rooms.

**Strengths:**

1. The paper is motivated by an important problem in traditional meta-RL approaches, where they maximize the cumulative returns across episodes rather than maximizing the final episodes, hence leading to insufficient exploration.

2. The authors proposed an interesting idea that uses the performance of the exploitation policy as a direct objective for the exploration policy to separate exploration and exploitation in meta-adaption.

**Weaknesses:**

1. The idea of separating exploration and exploitation in meta RL is not new. Several prior works, like MetaCURE[1] and DREAM [2], proposed similar ideas, but they are not properly discussed or compared.

2. The experiment is sufficient. The proposed approach is tested only on two tabular enironments, namely Meta-RL-Deceiving Bandits and Dark Treasure-Rooms. It is unclear how would the proposed method performance on more complex tasks like continuous control.

3. The writing of the method section can be improved. It needs proper formulas or an algorithm box to clearly demonstrate the methods for training the exploration and exploitation policies.


[1] Zhang, Jin, et al. "Metacure: Meta reinforcement learning with empowerment-driven exploration." International Conference on Machine Learning. PMLR, 2021.

[2] Liu, Evan Z., et al. "Decoupling exploration and exploitation for meta-reinforcement learning without sacrifices." International conference on machine learning. PMLR, 2021.

**Questions:**

1. Can you give explicit formulas of the objectives for training exploration and exploitation policies?

2. Can you explain why the proposed method takes so long to run on the tabular environments (50 GPU hours, according to Appendix C.1)?

3. How does the proposed perform on more complex tasks, like continuous control tasks (e.g., Meta-World)? How does the proposed method compare with MetaCURE and DREAM?

**Limitations:**

The authors discussed limitations of their work.

---

> ### Author Rebuttal · Authors · 2024-08-07
>
> Thank you for reading, engaging with, and critiquing our work. We greatly appreciate your time and feedback, which has significantly improved the paper. We are delighted that you consider First-Explore to be “motivated by an important problem” and “an interesting idea.”
>
> We have addressed your concerns below, and by doing so, we have substantially strengthened the paper. We hope you will consider our improvements and increase your score by 2 to 3 points. Without this score increase, the paper may not be selected for publication.
>
> # Weaknesses Response
> > [What about, for example, MetaCURE and DREAM? These methods separate exploration and exploitation]
>
> DREAM and MetaCURE are final-episode-reward meta-RL methods. In contrast, this paper and algorithm address cumulative-reward Meta-RL (see lines 41, 71, 126, 293).  An extensive discussion of MetaCure and DREAM is provided in Appendix F (see Section 2, lines 126-128, 'for the sake of completeness, we discuss final-episode-reward meta-RL and its connection to First-Explore in Appendix F'). Please see that Appendix for the answer to your question.
>
> To make it clearer, we will add text to Section 2 identifying DREAM and MetaCure  as final-episode-reward meta-RL methods and their difference to First-Explore.
> > [More complex environments are needed]
>
> We have added a more complex domain, Ray Maze (see the main response to all reviewers for details). This new domain is significantly more complex than the Dark Treasure Room; however, the same pattern of performance holds: First-Explore outperforms all controls, which fail catastrophically due to the deceptive nature of the domain.
>
> Furthermore, although the tabular environments are less complex than Ray Maze, the tabular results are highly informative. First-Explore identifies a challenge where state-of-the-art methods perform abysmally, often learning to remain stationary. Identifying this issue is a significant contribution of the paper, and is best demonstrated in the simplest environments possible (as is usually the case in science). It is surprising, and worth sharing with the NeurIPS community, how even simple environments can be unsolvable by existing state-of-the-art cumulative-reward meta-RL approaches.
>
> > [The methods section needs proper formulas or an algorithm box]
> > [What are the explicit formulas of the training objectives?]
>
> We have now included full pseudocode for the algorithm in the paper (see the main rebuttal response) and will edit the methods section for clarity.
> # Questions
> > [Why does First-Explore take so long to run on the tabular environments?]
>
> The hyperparameters and choice of architecture are not optimized for short meta-training durations. The transformer size, for example, prioritizes sufficient model capacity for any task over training efficiency on simpler tasks. It was also motivated by the desire to use a standard implementation (GPT2) of sufficient capacity to learn complex relationships. Similarly, the hyperparameter T (see line 184) enables exceptionally reliable training; however, it also increases meta-training time. If a shorter training time was desired, then these choices could be reconsidered, and future work could focus on optimizing First-Explore for such efficiency.
>
> However, the paper makes no claims on First-Explore’s meta-training efficiency. The paper identifies a new challenge with cumulative-reward meta-RL (the problem of needing to forgo rewards to explore), demonstrates the issue empirically, and provides a new method (First-Explore) that solves the challenge and achieves state-of-the-art results. We will summarize these details in the paper, including the exciting opportunity for future work to focus on substantially improving meta-training time.
> > How does First-Explore perform on more complex tasks (e.g., Meta-World)?
>
> We have added a new more complex domain - Ray Maze. First-Explore performs excellently in this domain (see the main reply for details).
>
> Meta-World’s environments do not require exploration that sacrifices rewards in early episodes, which is the central challenge this paper addresses. Evaluating First-Explore on such a benchmark would be interesting for future work, but it is beyond the scope of this paper.
> > How does the proposed method compare with MetaCURE and DREAM?
>
> MetaCURE and DREAM are final-episode-reward meta-RL methods designed for and applicable to the final-episode reward meta-RL setting. This paper exclusively makes claims about cumulative-reward meta-RL dynamics (see lines 41, 71, 126, 293, etc.). While comparing First-Explore to these (or any) methods is interesting, it is not essential to the scientific questions addressed here and is beyond this paper’s scope. That said, we do provide a discussion of this issue in Appendix F (please see it for the most direct answer to your question) and we speculate in the future works section that First-Explore may be applicable to the final-episode-reward setting.
>
> # Conclusion
> Thank you again for your feedback. We hope you will support a score that would lead to our work being shared with the NeurIPS community.

---

> > ### Comment · Reviewer_Vncg · 2024-08-14
> >
> > Thanks for the detailed comments. My concerns about the novelty part are resolved. However, I still find that the paper lacks an explicit mathematical formula for the objective of explorative and exploitive policies and that the computation cost can be too high. Considering the above, I have changed my score to 5.

---

### Author Rebuttal · Authors · 2024-08-07

Many thanks for reading our work. We are grateful for the feedback provided and for the time you spent engaging with and critiquing First-Explore. We are delighted that First-Explore addresses an "important" [Vncg], "highly relevant" [Erp8] problem "worth solving" [zDt1], and that it is an "interesting" [Vncg], "elegant" [zDt1] idea that provides "valuable insights" [Erp8]. Like you, we feel that First-Explore addresses (and identifies) an important problem of significant relevance to the meta-RL community, and that the paper has great potential to spark future works that build on it.

Using your feedback, we have strengthened the paper by adding a more challenging domain, Ray Maze, and by providing detailed training pseudocode (see below). We have also made several writing improvements. Please see our individual reviewer responses for replies to your questions and comments.

The current scores are low and prevent publication. In light of these substantial improvements, which we believe fully address your comments and concerns, we kindly request you consider raising your scores by 2-3 points to support sharing First-Explore with the NeurIPS community. We would appreciate it, and believe publishing this work will inform the community, improve understanding of RL, and inspire future research.

# Ray Maze

As requested, we have added a significantly more challenging domain, Ray Maze, demonstrating that First-Explore works beyond tabular and bandit environments (see rebuttal PDF for plots).

In Ray Maze, the agent must navigate a randomly generated maze of impassable walls, with the maze layout different for each sampled Ray Maze, to find one of three goal locations. Each goal location is a trap (70% probability, -1 penalty) or a treasure (30% probability, +1 reward). The agent can only receive one goal reward per episode. To perceive the maze, the agent receives 15 lidar observations (see PDF). Each lidar observation reports the distance to the nearest wall at a specific angle relative to the agent's orientation. It also details the orientation of the intercepted wall (east-west or north-south), and whether the lidar ray hit a goal location (but not whether it is a trap or a treasure). The agent has three actions: turn left, go forward, and turn right.

Ray Maze is a challenging domain for several reasons. It has a high-dimensional observation space (15 separate lidar measurements), complex action dynamics (with actions not commuting; for example, turning left then moving forward is different from moving forward then turning left), and a randomly generated maze that interacts with both movement and observation. Furthermore, the agent must learn from experience, and risk the traps in early episodes to enable exploiting and consistently finding treasure in later ones. Because goal locations are frequently traps, the agent can only obtain positive cumulative reward by first a) constantly searching for treasures in early episodes (despite this leading to negative expected reward) and then b) repeatedly exploiting in later episodes (navigating to identified treasures, while avoiding identified traps).

In this challenging domain, First-Explore significantly outperforms all other treatments (green lines are above all other lines in the rebuttal PDF), with First-Explore achieving over seven times the mean cumulative reward of the best control (0.46 vs. 0.06). The difference is statistically significant, with ($p < 6 \times 10^{-8}$) for all pairwise comparisons.

# Added Training Pseudocode
The cross-entropy loss calculation:
```python
def rollout(env, π, ψ, c_π, c_ψ):
	"""perform a single episode
	inputs: the environment (env),
	the agent policy π, the prediction policy ψ,
	and the current policies' contexts c_π, c_ψ
	returns the episode return, temp_loss, and updated contexts"""
	# n.b. temp_loss is only used if the episode meets a condition
	# see (*) in the conditional_action_loss function
	temp_loss, r = 0, 0
	s = env.reset() # state s
	for i in range(env.episode_length):
		# calculate action probabilities p_π, p_ψ for both policies
		# and update context
		p_π, c_π = π.action_probabilities(s, c_π)
		p_ψ, c_ψ = ψ.action_probabilities(s, c_ψ)
		a = sample_action(p_π)
		temp_loss += cross_entropy(a, p_π * p_ψ) # hardamard product
		# * p_π ensures action diversity by weighting against likely actions
		s = env.step(s, a)
		r += s.reward
	return r, temp_loss, c_π, c_ψ

def conditional_action_loss(φ, θ, D, b):
	"""calculates First-Explore loss for both exploring and exploiting on domain D
	using the agent and predictor parameters φ, θ
	and baseline reward b"""
	env = sample_env(D)
	π_explore, π_exploit = load_policies(θ)
	ψ_explore, ψ_exploit = load_policies(φ)
	c_π, c_ψ = set(), set() # the agent and predictor contexts
	loss, best_r = 0, b
	for i in range(D.episode_num):
        r_explore, l_explore, c_π, c_ψ = rollout(env, π_explore,
    	                                         ψ_explore, c_π, c_ψ)
    	r_exploit, l_exploit, _, _ = rollout(env, π_exploit, ψ_exploit, c_π, c_ψ)
        # ^exploit context not kept
    	# (*) accumulate loss if:
    	if r_exploit > best_r: # explore episode is 'maximal'
        	loss += l_explore
    	if r_exploit >= best_r: # exploit episode is 'informative'
        	loss += l_exploit
    	if r_exploit > best_r:
        	best_r = r_exploit
	return loss
```

Training with the above loss:
```python
def train(epoch_num, T, D, b):
	"""example First-Explore training (ignoring batchsize)
    runs the meta-rollouts, accumulating a loss
	this loss is then auto-differentiated"""
	T_counter = 0
	φ = θ = init_params()
	for i in range(epoch_num):
    	# θ is the agent behavior parameters
    	# φ is the prediction parameter (for double-DQN-style updates)
    	Δφ = ∂/∂φ(conditional_action_loss(φ, θ, D, b))
    	φ -= step_size * Δφ
    	# Update θ every T epochs
    	T_counter += 1
    	if T_counter == T:
        	θ = φ
        	T_counter = 0
	return θ
```

---

### Decision · Program_Chairs · 2024-09-25

**Decision:**

Accept (poster)

**Comment:**

The paper presents an interesting and valuable contribution to the Meta-RL field by addressing the challenge of balancing exploration and exploitation, particularly in the presence of deceptive rewards.
However, the concerns regarding mathematical clarity, computational cost, and the limited complexity of the tested environments suggest that there is still room for improvement. The paper would benefit from further refinement, particularly in the explicit formulation of its objectives and broader experimentation.
Given the overall positive trajectory of the reviews post-rebuttal, I recommend accepting this paper.
The authors need to incorporate all the improvements proposed during the rebuttals and all the suggestions provided by the reviewers.